# Differentially Private High-dimensional Variable Selection via Integer Programming

**Petros Prastakos**
Operations Research Center
MIT
Cambridge, MA 02139, USA
pprastak@mit.edu

**Kayhan Behdin**
LinkedIn
Sunnyvale, CA 94085, USA
kbehdin@linkedin.com

**Rahul Mazumder**
Operations Research Center
Sloan School of Management
MIT
Cambridge, MA 02139, USA
rahulmaz@mit.edu

## Abstract

Sparse variable selection improves interpretability and generalization in high-dimensional learning by selecting a small subset of informative features. Recent advances in Mixed Integer Programming (MIP) have enabled solving large-scale non-private sparse regression—known as Best Subset Selection (BSS)—with millions of variables in minutes. However, extending these algorithmic advances to the setting of Differential Privacy (DP) has remained largely unexplored. In this paper, we introduce two new pure differentially private estimators for sparse variable selection, levering modern MIP techniques. Our framework is general and applies broadly to problems like sparse regression or classification, and we provide theoretical support recovery guarantees in the case of BSS. Inspired by the exponential mechanism, we develop structured sampling procedures that efficiently explore the non-convex objective landscape, avoiding the exhaustive combinatorial search in the exponential mechanism. We complement our theoretical findings with extensive numerical experiments, using both least squares and hinge loss for our objective function, and demonstrate that our methods achieve state-of-the-art empirical support recovery, outperforming competing algorithms in settings with up to $p = 10^4$. Code is available at https://github.com/petrosprastakos/DP-variable-selection.

## 1 Introduction

High-dimensional datasets are increasingly common, but extracting meaningful models is challenging due to overfitting and lack of interpretability. Statistical regularizations that encourage model simplicity from certain perspectives have been successful in addressing such challenges, becoming a staple of high-dimensional statistics and machine learning. One such common regularization is sparsity [16, 15], where one seeks to choose a small subset of features in the data to form the statistical model.

In this paper, we focus on the problem of sparse variable selection. Given the data matrix $\boldsymbol{X} \in \mathbb{R}^{n \times p}$ and the observations $\boldsymbol{y} \in \mathbb{R}^n$, we seek to obtain an estimator $\boldsymbol{\beta}$ that describes the data well with only a few coordinates of $\boldsymbol{\beta}$ being nonzero. A natural first formulation for this problem is

$$\min_{\boldsymbol{\beta} \in \mathbb{R}^p} \sum_{i=1}^{n} \ell(y_i, \boldsymbol{x}_i^T \boldsymbol{\beta}) \ \text{ s.t. } \ \|\boldsymbol{\beta}\|_0 \leq s, \ \|\boldsymbol{\beta}\|_2^2 \leq r^2 \tag{1}$$

where $\| \cdot \|_0$ counts the number of nonzero coordinates of a vector. In the case where $\ell(y_i, \boldsymbol{x}_i^T \boldsymbol{\beta}) = (y_i - \boldsymbol{x}_i^T \boldsymbol{\beta})^2$, the objective becomes least squares, and the problem is referred to as Best Subset Selection (BSS, Miller [27]). The constraint $\|\boldsymbol{\beta}\|_0 \leq s$ enforces sparsity via the sparsity budget

39th Conference on Neural Information Processing Systems (NeurIPS 2025).

$s > 0$, and the constraint $\|\boldsymbol{\beta}\|_2^2 \leq r^2$ for some $r > 0$ serves as an additional (ridge) regularization. A sparse linear estimator can be more interpretable and have better statistical performance [15, 16, 34].

Real-world datasets often contain confidential and personal information, that should be protected. Hence, recent years have seen a surge in private learning algorithms, hoping to preserve sensitive information while extracting useful statistical knowledge. In particular, Differential Privacy (DP, Dwork [9]) has garnered significant interest in the machine learning and statistics literature. On a high level, DP aims to ensure one cannot obtain too much information from the private dataset, via querying the statistical model in an adversarial way. A significant body of work is dedicated to designing DP algorithms for general machine learning tasks [25, 10, 14, 11, 12], as well as specialized algorithms for specific statistical problems. Particularly, there is a long line of work studying the sparse linear regression problem [31, 32, 23, 24, 29].

In this paper, we develop two scalable pure DP algorithms for variable selection under a broad framework where one releases the optimal support in (1) (i.e., the location of nonzero coordinates in the optimal $\boldsymbol{\beta}$). To our knowledge, we are the first to incorporate MIP techniques for this task. While our support recovery results are derived specifically for the Best Subset Selection (BSS) setting, we provide pure DP guarantees for our methods that hold for general loss functions (not just least squares). Specifically, we make the following contributions:

1. Our first method, named **top-R**, satisfies pure DP under only a standard boundedness assumption on the data (achievable via clipping). For BSS, it achieves support recovery with high probability whenever $\beta_{\min} := \min_{j \in \{i:\beta_i^* \neq 0\}} |\beta_j^*| \gtrsim \sqrt{\max\{1, s^2/\epsilon\}(\log p)/n}$, matching the non-private minimax-optimal $\sqrt{(\log p)/n}$ threshold in the low-privacy regime.

2. Our second method, named **mistakes**, also satisfies pure DP, but requires an additional separation assumption on the objective gap for pure-DP guarantees. In BSS, this condition holds with high probability under $\beta_{\min} \gtrsim \sqrt{s \log p/n}$, with the milder condition for support recovery of $\beta_{\min} \gtrsim \sqrt{\max\{1, 1/\epsilon\}(s \log p)/n}$, which aligns with the condition of [29] in the high-privacy regime.

3. Empirically, our methods outperform the other DP variable selection methods in the literature, including the state-of-the-art approximate DP MCMC approach of [29] for BSS, under a wide range of parameter values and up to $p = 10{,}000$. We also show strong empirical performance in wider settings, including sparse classification with hinge loss. Our results demonstrate that DP variable selection with provable guarantees and practical scalability is possible by combining optimization and privacy.

## 1.1 Related Work

**DP variable selection.** Most of the existing DP literature focuses on non-sparse linear regression, or $\ell_2$ risk excess in sparse regression [31, 33, 21, 7, 22]. For the specific problem of DP variable selection, previous works have focused on the sparse regression setting. As Lasso tends to promote sparsity, an interesting line of work is based on releasing the variables selected by Lasso in a private fashion [32, 23]. [32] introduce two propose-test-release algorithms for variable selection. However, the failure probability for support recovery for these methods does not approach 0 with growing sample size. [23] propose a computationally efficient resample-and-aggregate [28] algorithm, which underperforms compared to our methods in practice, and requires a stronger $\beta_{\min}$ condition than in our methods in the case of BSS. Lei et al. [24] propose an algorithm based on the exponential mechanism, requiring to enumerate all feasible supports in (1), limiting the scalability of their method. Recently, Roy and Tewari [29] have proposed a new method based on the notion of Markov chain mixing to obtain approximate DP solutions for BSS, resulting in a statistically strong estimator. While our $\beta_{\min}$ conditions are comparable with theirs in the low-privacy regime of top-$R$ or high-privacy regime of mistakes, we note that we achieve pure-DP guarantees for general loss functions, our algorithms have scope beyond BSS, and our empirical performance is stronger across a broad range of parameters.

**Modifications to the exponential mechanism.** The methods we introduce in this paper involve modifications to the exponential mechanism, a fundamental DP algorithm, in order to reduce the outcome set of our sampling distribution. Some other truncations of the exponential mechanism have existed in the DP literature. First, the Restricted Exponential Mechanism (REM) [6] for private

mean estimation samples from the exponential mechanism restricted to points of sufficiently large Tukey depth, together with a private "safety" check that the restricted set is well behaved. Second, the Truncated Exponential Mechanism (TEM) for metric-DP on text [30] restricts selection to a $\gamma$-ball around the input and collapses the remainder of the domain into a single $\perp$ bucket—equivalently, assigning the outside set a shared score. While the spirit of truncation is analogous to the modifications proposed in this paper, our methods target combinatorial support selection under pure DP, rather than metric-DP over text data or mean estimation.

**Notation.** We let $[p] = \{1, \cdots, p\}$. Data points follow $(\boldsymbol{x}_i, y_i) \in \mathcal{Z} = \mathcal{X} \times \mathcal{Y} \subset \mathbb{R}^p \times \mathbb{R}$, with $\mathcal{D} = (\boldsymbol{X}, \boldsymbol{y}) \in \mathcal{Z}^n$ for a dataset containing $n$ observations.

## 2 Method

**Background on Differential Privacy**  Before continuing with our selection procedure, let us formalize the notion of differential privacy.

**Definition 1** ([9])**.** Given the privacy parameters $(\varepsilon, \delta) \in \mathbb{R}^+ \times \mathbb{R}^+$, a randomized algorithm $\mathcal{A}(\cdot)$ is said to satisfy the $(\varepsilon, \delta)$-DP property if

$$\mathbb{P}(\mathcal{A}(\mathcal{D}) \in K) \leq e^\varepsilon \mathbb{P}(\mathcal{A}(\mathcal{D}') \in K) + \delta$$

for any measurable event $K \subset \text{range}(\mathcal{A})$ and for any pair of neighboring datasets $\mathcal{D}$ and $\mathcal{D}'$.

We note that in Definition 1, the probability is taken over the randomness of the algorithm $\mathcal{A}$. When $\delta > 0$, the $(\varepsilon, \delta)$-DP property is also commonly referred to as *approximate* differential privacy, while the special case where $\delta = 0$ is commonly referred to as *pure* differential privacy.

Next, let us briefly review the exponential mechanism [25], a general mechanism to achieve pure DP. Consider a general task where the dataset $\mathcal{D} \in \mathcal{Z}^n$ is given, and we seek to design a procedure such as $\mathcal{A} : \mathcal{Z}^n \to \mathcal{O}$ to choose the outcome of the task, where $\mathcal{O}$ is the set of possible outcomes. We also assume we are given an objective function such as $\mathcal{R} : \mathcal{O} \times \mathcal{Z}^n \to \mathbb{R}$, where a smaller objective indicates a more desirable outcome. The global sensitivity of the objective is then defined as

$$\Delta = \max_{o \in \mathcal{O}} \max_{\substack{\mathcal{D}, \mathcal{D}' \in \mathcal{Z}^n \\ \mathcal{D}, \mathcal{D}' \text{ are neighbors}}} \mathcal{R}(o, \mathcal{D}) - \mathcal{R}(o, \mathcal{D}'). \tag{2}$$

**Lemma 1** (Exponential Mechanism, McSherry and Talwar [25])**.** *The exponential mechanism $\mathcal{A}_E(\cdot)$ that follows*

$$\mathbb{P}(\mathcal{A}_E(\mathcal{D}) = o) \propto \exp\left(-\frac{\varepsilon \mathcal{R}(o, \mathcal{D})}{2\Delta}\right), \quad \forall o \in \mathcal{O} \tag{3}$$

*ensures $(\varepsilon, 0)$-DP.*

### 2.1 Selection Procedure

#### 2.1.1 Top-$R$ Method

The main inspiration for our selection procedure is the exponential mechanism, defined in Lemma 1. In particular, in the BSS problem we seek to select a subset of features with size $s$ that are a good linear predictor of $n$ observations $\boldsymbol{y}$. Therefore, a natural choice for the outcome set in BSS is the set of all subsets of $[p]$ with size $s$, $\mathcal{O} = \{S \subseteq [p] : |S| = s\}$. Next, a natural choice for the objective in the BSS problem for each $S$ is the least squares loss, when the regression coefficients can only be nonzero for features in $S$. Formally,

$$\mathcal{R}(S, \mathcal{D}) = \min_{\boldsymbol{\beta} \in \mathbb{R}^{|S|}} \sum_{i=1}^n \ell(y_i, (\boldsymbol{x}_i)_S^T \boldsymbol{\beta}) \text{ s.t. } \|\boldsymbol{\beta}\|_2^2 \leq r^2 \tag{4}$$

where $(\boldsymbol{x}_i)_S$ is the vector $\boldsymbol{x}_i$ with columns indexed by $S$.

Note that, if the elements of the dataset $\mathcal{D}$ are unbounded, the global sensitivity of $\mathcal{R}(S, \mathcal{D})$ may be unbounded. We thus make the following boundedness assumption on $\mathcal{D}$.

**Assumption 1.** There exist positive constants $b_y, b_x$ such that $\sup_{y \in \mathcal{Y}} |y| \leq b_y, \sup_{\boldsymbol{x} \in \mathcal{X}} \|\boldsymbol{x}\|_\infty \leq b_x$.

We note that, in practice, one might not know the exact values of $b_x, b_y$, or such values might not exist. In such cases, one can clip the values of $\boldsymbol{X}, \boldsymbol{y}$ to satisfy the boundedness requirements of Lemma A.1. In Section A.1 in the appendix, we show that, for the special case where our objective function is least squares, assumption 1 yields $\Delta \leq 2b_y^2 + 2b_x^2 r^2 s$.

**Our Proposal**  With a bounded global sensitivity, one can directly apply the exponential mechanism on $\mathcal{R}(S, \mathcal{D})$ and achieve a $(\varepsilon, 0)$-DP procedure for problem 1. The difficulty in variable selection under DP constraints arises from the need to enumerate all feasible solutions in $\mathcal{O}$. However, one can argue that if a support $S$ is far from the optimal one, the least-squares objective $\mathcal{R}(S, \mathcal{D})$ is likely to be large, therefore, the probability mass of $S$ in (3) should be small. Therefore, one might ask:

> ***Is it necessary to have access to*** $\mathcal{R}(S, \mathcal{D})$ ***for all*** $S \in \mathcal{O}$ ***in the exponential mechanism?***

Specifically, for the moment, suppose we have access to an oracle that for a fixed $R > 1$, can return $R$ feasible supports from $\mathcal{O}$ that have the smallest objectives. Formally, assume we can access $\hat{S}_1(\mathcal{D}), \cdots, \hat{S}_R(\mathcal{D})$ where

$$\hat{S}_k(\mathcal{D}) \in \arg\min_S \mathcal{R}(S, \mathcal{D}) \text{ s.t. } S \subseteq [p], \ |S| = s, \ S \neq \hat{S}_i(\mathcal{D}), i = 1, \cdots, k-1. \tag{5}$$

In particular, $\hat{S}_1(\mathcal{D})$ is the optimal support for BSS in (1). Then, based on our discussion above, if $R$ is sufficiently large, the values $\mathcal{R}(\hat{S}_k(\mathcal{D}), \mathcal{D})$ for $k \geq R$ are expected to be significantly larger than $\mathcal{R}(\hat{S}_k(\mathcal{D}), \mathcal{D})$ for $k \ll R$. Therefore, most of the probability mass of the distribution in (3) is concentrated around $\hat{S}_k(\mathcal{D})$ for $k \ll R$. Hence, we might not need to have access to the exact values of $\mathcal{R}(\hat{S}_k(\mathcal{D}), \mathcal{D})$ for $k \geq R$, as long as we can replace them with a suitable lower bound. This lower bound can be taken as $\mathcal{R}(\hat{S}_R(\mathcal{D}), \mathcal{D})$. To this end, we propose the sampling procedure $\hat{\mathcal{M}}$, shown as Algorithm 1 below, where $\mathbb{P}_0$ is the probability distribution following

$$\mathbb{P}_0(k) \propto \begin{cases} \exp\left(-\varepsilon \mathcal{R}(\hat{S}_k(\mathcal{D}), \mathcal{D})/(2\Delta)\right) & \text{if } k \leq R \\ \left(\binom{p}{s} - R\right) \exp\left(-\varepsilon \mathcal{R}(\hat{S}_R(\mathcal{D}), \mathcal{D})/(2\Delta)\right) & \text{if } k = R + 1. \end{cases} \tag{6}$$

---

**Algorithm 1** Top-$R$ method

---

1: **procedure** $\hat{\mathcal{M}}(\mathcal{D}, s, b_x, b_y, r, R, T)$
2:     Clip $\boldsymbol{X}, \boldsymbol{y}$ to $b_x, b_y$, respectively, as in Lemma A.1. Take $\Delta$ as in Lemma A.1. Form $\mathbb{P}_0$ in (6).
3:     Draw $a(\mathcal{D}) \sim \mathbb{P}_0$
4:     **if** $a(\mathcal{D}) \leq R$ **then**
5:         **return** $\hat{S}_{a(\mathcal{D})}(\mathcal{D})$
6:     **else**
7:         **return** $\mathcal{M}_0(\mathcal{D}, R, T)$
8: **procedure** $\mathcal{M}_0(\mathcal{D}, R, T)$
9:     **for** $t \leq T$ **do**
10:        Draw $S \in \mathcal{O}$ uniformly at random, independent of $\mathbb{P}_0$.
11:        **if** $S \in \{\hat{S}_k(\mathcal{D}), k > R\}$ **then**
12:           Break
13:     **return** $S$

---

Intuitively speaking, $\hat{\mathcal{M}}$ replaces $\mathcal{R}(\hat{S}_k(\mathcal{D}), \mathcal{D})$ for $k \geq R$ with $\mathcal{R}(\hat{S}_R(\mathcal{D}), \mathcal{D})$ and then "approximately" samples from the exponential mechanism. To this end, let

$$\hat{\mathcal{R}}(S, \mathcal{D}) = \begin{cases} \mathcal{R}(S, \mathcal{D}) & \text{if } S \in \{\hat{S}_1(\mathcal{D}), \cdots, \hat{S}_R(\mathcal{D})\} \\ \mathcal{R}(\hat{S}_R(\mathcal{D}), \mathcal{D}) & \text{otherwise} \end{cases} \tag{7}$$

where we substitute $\mathcal{R}(\hat{S}_k(\mathcal{D}), \mathcal{D})$ for $k \geq R$ with $\mathcal{R}(\hat{S}_R(\mathcal{D}), \mathcal{D})$. Suppose $\hat{\mathcal{A}}_E$ is the exponential mechanism that uses the objective $\hat{\mathcal{R}}$. If $a(\mathcal{D}) \leq R$ in Algorithm 1, we return $\hat{S}_{a(\mathcal{D})}(\mathcal{D})$. Note that $\mathbb{P}(\hat{\mathcal{A}}_E(\mathcal{D}) = \hat{S}_{a(\mathcal{D})}(\mathcal{D})) = \mathbb{P}_0(a(\mathcal{D}))$ in this case, showing $\hat{\mathcal{M}}$ mimics the exponential mechanism

$\hat{\mathcal{A}}_E$. If $a(\mathcal{D}) = R+1$, to mimic $\hat{\mathcal{A}}_E$, we have to sample uniformly from the set $\mathbb{S} = \{\hat{S}_k(\mathcal{D}), k > R\}$ as $\mathbb{P}(\hat{\mathcal{A}}_E)$ is uniform on $\mathbb{S}$, by the definition of $\hat{\mathcal{R}}$ in (7). However, $\mathbb{S}$ is exponentially large in general. Therefore, we invoke $\mathcal{M}_0$ that in the limit of $T \to \infty$, samples uniformly from $\mathbb{S}$.

Observe that, in the case where $R = \binom{p}{s}$, we have that the distribution $\mathbb{P}_0$ is the same as the exponential mechanism that uses objective $\mathcal{R}$ as in [29], which is $(\varepsilon, 0)$-DP by Lemma 1. Below, we show this procedure satisfies pure DP for any $R \in \{2, ..., \binom{p}{s} - 1\}$ as well. We defer all proofs to the appendix.

**Theorem 1** (Privacy for top-$R$ method). *Suppose $T > 1$, $1 < R < \binom{p}{s}$, and that assumption 1 holds. The procedure $\hat{\mathcal{M}}$ in Algorithm 1 is $(\varepsilon', 0)$-DP where*

$$\varepsilon' = \log\left(e^\varepsilon + \frac{q^T}{\delta_0}\right) - \log\left(1 - q^T\right), \quad \delta_0 = \frac{\exp(-n\varepsilon b_y^2/(2\Delta))}{\binom{p}{s}}, \quad q = \frac{R}{\binom{p}{s}}.$$

*In particular, if $T = \infty$, the procedure $\hat{\mathcal{M}}$ is $(\varepsilon, 0)$-DP.*

Theorem 1 shows that, regardless of the choice of $R$, as $T \to \infty$, we have that $\epsilon' \to \epsilon$. However, we note that $\epsilon'$ *increases* with $R$, so there is more privacy loss with increasing $R$, as we have that

$$\frac{\partial\left(\log\left(e^\varepsilon + \frac{q^T}{\delta_0}\right) - \log\left(1 - q^T\right)\right)}{\partial q} = T\, q^{T-1}\left[\frac{1}{\delta_0\left(e^\varepsilon + q^T/\delta_0\right)} + \frac{1}{1 - q^T}\right] > 0.$$

The privacy loss is in contrast to the effect on accuracy, as we note that a larger $R$ in Algorithm 1 should intuitively lead to better support recovery. We formalize this intuition in Lemma A.5 in the appendix. For $k > R$, we underestimate $\mathcal{R}(\hat{S}_k, \mathcal{D})$ with $\mathcal{R}(\hat{S}_R, \mathcal{D})$, consequently increasing the probability mass given to supports $\hat{S}_k$ in the procedure $\hat{\mathcal{A}}_E$. This reduces the probability mass for the best support $\hat{S}_1$. Therefore, in practice, we like to choose a larger $R$ to explore the objective landscape better, however, a very large $R$ can make the computation slower.

The sampling procedure in Algorithm 1 only requires sampling from $\mathbb{P}_0$ (which is supported on $R + 1$ different values), and sampling sparse supports from a uniform distribution (in procedure $\mathcal{M}_0$), which can be done efficiently. Therefore, this procedure circumvents the need to sample from a non-uniform distribution with exponentially large support. Importantly, Algorithm 1 satisfies pure $(\varepsilon', 0)$-DP, with $\epsilon' = \epsilon$ as $T \to \infty$. To our knowledge, no such algorithm exists for BSS that can scale to problems with tens of thousands of variables.

### 2.1.2 Mistakes Method

Our second proposed mechanism assigns probabilities based on the number of mistakes from the optimal solution. Namely, we define $\tilde{S}_0(\mathcal{D}) = \hat{S}_1(\mathcal{D}) = \arg\min_S \mathcal{R}(S, \mathcal{D})$, and then we proceed to partition the $\binom{p}{s} - 1$ supports based on the number of mistakes from $\tilde{S}_0(\mathcal{D})$. We denote the partition $P_1(\mathcal{D}), P_2(\mathcal{D}), ..., P_s(\mathcal{D})$. Let $P_0(\mathcal{D}) = \{\tilde{S}_0(\mathcal{D})\}$. We then have that for $i \in [s]$

$$\tilde{S}_i(\mathcal{D}) = \arg\min_{S \in P_i(\mathcal{D})} \mathcal{R}(S, \mathcal{D}). \tag{8}$$

Our mistakes method, denoted $\tilde{\mathcal{M}}$, assigns probabilities according to the element of the partition that a support belongs to. Namely, if $S \in P_k(\mathcal{D})$ for $k \in \{0, 1, ..., s\}$, we have that

$$\mathbb{P}[\tilde{\mathcal{M}}(\mathcal{D}) = S] = \frac{\exp(\frac{-\epsilon\mathcal{R}(\tilde{S}_k(\mathcal{D}),\mathcal{D})}{2\Delta})}{\sum_{i=0}^s |P_i(\mathcal{D})| \exp(\frac{-\epsilon\mathcal{R}(\tilde{S}_i(\mathcal{D}),\mathcal{D})}{2\Delta})} = \frac{\exp(\frac{-\epsilon\mathcal{R}(\tilde{S}_k(\mathcal{D}),\mathcal{D})}{2\Delta})}{\sum_{i=0}^s \binom{p-s}{i}\binom{s}{i} \exp(\frac{-\epsilon\mathcal{R}(\tilde{S}_i(\mathcal{D}),\mathcal{D})}{2\Delta})}$$

For $S \in P_k(\mathcal{D})$, define $\tilde{\mathcal{R}}(S, \mathcal{D}) = \mathcal{R}(\tilde{S}_k(\mathcal{D}), \mathcal{D})$.

Below, we show this method is $(\epsilon, 0)$-DP under a lower bound assumption on the gap in objective value between $\hat{S}_1(\mathcal{D})$ and $\hat{S}_2(\mathcal{D})$.

**Theorem 2** (Privacy for mistakes method). *Suppose assumption 1 holds and that $\mathcal{R}(\hat{S}_2(\mathcal{D}), \mathcal{D}) - \mathcal{R}(\hat{S}_1(\mathcal{D}), \mathcal{D}) > 2\Delta$. Then, the mistakes method is $(\epsilon, 0)$-differentially private.*

We note that, unlike the privacy of our top-$R$ method in Theorem 1, which requires no additional assumptions aside from assumption 1, Theorem 2 requires stronger conditions for the privacy of the mistakes method. In Lemma A.6 in the appendix, we show that, under the sufficient condition that $\tau \gtrsim \frac{s \log p}{n}$, where $\tau$ is defined in the following section, and the additional assumptions 2-4, we have that the inequality $\mathcal{R}(\hat{S}_2(\mathcal{D}), \mathcal{D}) - \mathcal{R}(\hat{S}_1(\mathcal{D}), \mathcal{D}) > 2\Delta$ holds with high probability.

**Remark 1.** While the privacy of our mistakes method relies on an additional assumption that occurs with high probability (as shown in Lemma A.6 in the appendix), providing a privacy guarantee with additional assumptions is not uncommon in the literature. For example, the privacy guarantees of [29], which is the closest competitor to our method, depends on assumptions that hold with high probability. More specifically, the privacy proof of the Markov Chain Monte Carlo (MCMC) algorithm in [29] relies on the assumption that the mixing of the Markov Chain used for sampling with its stationary distribution has happened. However, this mixing can only be guaranteed with high probability, and under additional assumptions on the underlying model—see Theorem 4.3 of [29] for more details. In contrast, our top-R method is always private (assuming $b_x, b_y$ are finite), and our mistakes method is private under assumptions that are similar to the ones in Theorem 4.3 of [29].

As another example, [32] uses the stability of Lasso, to present a DP method for support recovery in sparse linear regression. However, the stability of Lasso only holds under certain assumptions on the data, such as the boundedness of the noise and restricted strong convexity. Such assumptions might only hold with high probability in practice, resulting in privacy guarantees that hold with high probability. For more details, we refer to Theorem 8 of [32].

## 3 Statistical Theory

For the theoretical results in this section, we focus on the setting of BSS. Consider the model

$$y = X\beta^* + \epsilon$$

where $\{\epsilon_i\}_{i \in [n]}$ are i.i.d. zero-mean sub-Gaussian random variables with parameter $\sigma$, and the feature vector $\beta^*$ is unknown but is assumed to be $s$-sparse (i.e. its support size $|S^*| = |\{i : \beta_i^* \neq 0\}| = s \ll p$). In the remainder of this section, we provide sufficient conditions for our proposed methods to recover $S^*$ with high probability. We first state our additional assumptions.

**Assumption 2.** There exists positive constant $M$ such that $\|\beta^*\|_2 \leq M$.

**Assumption 3.** There exist positive constants $\kappa_-, \kappa_+$ such that, for all $S$ such that $|S| = s$, we have

$$\kappa_- \leq \lambda_{\min}(X_S^\top X_S/n) \leq \lambda_{\max}(X_S^\top X_S/n) \leq \kappa_+.$$

**Assumption 4.** The sparsity level $s$ follows the inequality $s \leq n/\log p$, and $p \geq 3$.

Assumption 2 tells that the true parameter $\beta^*$ lies inside an $\ell_2$ ball. Similar boundedness assumptions are fairly standard in the DP literature [35, 24, 7]. Assumption 3 is the Sparse Riesz Condition (SRC), which is a well-known assumption in the high-dimensional statistics literature [36, 20, 26]. Finally, Assumption 4 essentially assumes that $s = o(n)$, i.e., sparsity grows slowly relative to sample size.

Define the set of supports that make $t \in [s]$ mistakes from the true support as

$$\mathcal{A}_t = \{S \subset [p] : |S| = s, |S \setminus S^*| = t\}.$$

Let $\hat{\Sigma} = n^{-1}X^T X$ be the sample covariance and $\hat{\Sigma}_{S_1,S_2}$ be the submatrix of $\hat{\Sigma}$ with row indices in $S_1$ and column indices in $S_2$. Let $P_{X_S} = X_S(X_S^T X_S)^{-1}X_S^T$ denote the projection to the column space of $X_S$. Then we have that

$$y = X_{S^*}\beta_{S^*}^* + \epsilon = P_{X_S}X_{S^*}\beta_{S^*}^* + (I_n - P_{X_S})X_{S^*}\beta_{S^*}^* + \epsilon$$

and $(I_n - P_{X_S})X_{S^*}\beta_{S^*}^*$ describes the part of the signal that cannot be linearly explained by $X_S$. Define also

$$\hat{D}(S) = \hat{\Sigma}_{S^*\setminus S, S^*\setminus S} - \hat{\Sigma}_{S^*\setminus S, S}\hat{\Sigma}_{S,S}^{-1}\hat{\Sigma}_{S, S^*\setminus S}.$$

which is the covariance of the residuals of $X_{S^*\setminus S}$ after being regressed on $X_S$. We have that $\frac{1}{n}\left\|(I_n - P_{X_S})X_{S^*\setminus S}\beta_{S^*\setminus S}^*\right\|_2^2 = \beta_{S^*\setminus S}^{*\top}\hat{D}(S)\beta_{S^*\setminus S}^*$ which we can intuitively consider as the

discrimination margin between $S$ and the true support $S^*$. The larger this quantity, the easier it is for BSS to discriminate between $S^*$ and any other candidate model $S$.

We now introduce the central quantity of interest in analyzing support recovery, the *identifiability margin*, defined as

$$\tau = \min_{S \in \cup_{t=1}^{s} \mathcal{A}_t} \frac{\boldsymbol{\beta}_{S^* \setminus S}^{* \top} \hat{\boldsymbol{D}}(S) \boldsymbol{\beta}_{S^* \setminus S}^*}{|S^* \setminus S|}.$$

We observe that, the more correlated the features, the closer $\tau$ is to 0, and harder it is for BSS to distinguish between the true model and any other candidate support, so exact support recovery is harder. If the features are more uncorrelated, $\tau$ increases so exact support recovery is easier.

This intuition is made rigorous in Theorem 2.1 of [13], where it is shown that $\tau \gtrsim \frac{\log p}{n}$ is a sufficient condition to have

$$\{S^*\} = \arg \min_{S \in \mathcal{O}} \mathcal{R}_{ols}(S, \mathcal{D}), \quad \text{where} \quad \mathcal{R}_{ols}(S, \mathcal{D}) = \min_{\boldsymbol{\beta} \in \mathbb{R}^s} \|\boldsymbol{y} - \boldsymbol{X}_S \boldsymbol{\beta}\|_2^2$$

with high probability, i.e. to have $S^*$ be the unique minimizer for the BSS problem (with unconstrained $\ell_2$ norm on $\boldsymbol{\beta}$) with high probability. Such a theorem offers us support recovery guarantees for the non-private $\ell_0$-sparse ordinary least squares problem.

We now transition to the private setting of the $\ell_2$-constrained version of BSS, and offer support guarantees for our proposed methods under this setting. In the following theorem, we provide sufficient conditions for our private top-$R$ method to recover the true support with high probability.

**Theorem 3** (Support recovery for top-$R$ method)**.** *Suppose that assumptions 1-4 hold. Set $r \geq (\frac{\kappa_+}{\kappa_-})M + 4\frac{\sigma b_x}{\kappa_-}$. Set $\Delta = 2b_y^2 + 2b_x^2 r^2 s$. Then, there exists universal constant $C > 0$ such that, whenever*

$$\tau \geq \max\{C\sigma^2, \frac{8\Delta}{\epsilon}s\}\frac{\log p}{n},$$

*we have that*

$$\mathbb{P}(\hat{\mathcal{M}}(\mathcal{D}) = S^*) \geq \frac{1 - 10sp^{-2}}{1 + p^{-s}}.$$

**Comparison with previous work:** Theorem 3 shows that, using the appropriate global sensitivity bound and lower bound on $r$, a sufficient condition for recovering the true support with high probability is $\tau \gtrsim \max\{\frac{\log p}{n}, \frac{s^2 \log p}{n\epsilon}\}$, compared to $\tau \gtrsim \max\{\frac{\log p}{n}, \frac{s \log p}{n\epsilon}\}$ for the exponential mechanism applied to $\mathcal{R}(S, \mathcal{D})$ as in Theorem 3.5 of [29]. Observe that, in a low privacy regime, the $\frac{\log p}{n}$ term dominates, aligning with the [13] sufficient condition in the non-private setting. The extra factor of $s$ in the second term of our condition is expected, as we are not making any additional assumptions on the choice of $R$ or on the number of mistakes of the enumerated supports.

In the following theorem, we provide *weaker* sufficient conditions for our private mistakes method to recover the true support with high probability.

**Theorem 4** (Support recovery for mistakes method)**.** *Suppose that assumptions 1-4 hold. Set $r \geq (\frac{\kappa_+}{\kappa_-})M + 4\frac{\sigma b_x}{\kappa_-}$. Set $\Delta = 2b_y^2 + 2b_x^2 r^2 s$. Then, there exists a universal constant $C > 0$ such that, whenever*

$$\tau \geq \max\{C\sigma^2 s, \frac{16\Delta}{\epsilon}\}\frac{\log p}{n},$$

*we have that*

$$\mathbb{P}(\tilde{\mathcal{M}}(\mathcal{D}) = S^*) \geq \frac{1 - 18sp^{-2}}{1 + 2p^{-2}}.$$

**Comparison with previous work:** Theorem 4 shows that, using the appropriate global sensitivity bound and lower bound on $r$, a sufficient condition for recovering the true support with high probability is $\tau \gtrsim \max\{\frac{s \log p}{n}, \frac{s \log p}{n\epsilon}\}$, compared to $\tau \gtrsim \max\{\frac{\log p}{n}, \frac{s \log p}{n\epsilon}\}$ for the exponential mechanism applied to $\mathcal{R}(S, \mathcal{D})$ as in Theorem 3.5 of [29]. Our condition thus matches that in their paper for high privacy regimes. The strength in our result lies in noting that, unlike the exponential mechanism in [29], which requires access to $\mathcal{R}(S, \mathcal{D})$ for all $\binom{p}{s}$ supports $S \subset [p]$ such that $|S| = s$, our method only requires access to the $s + 1$ supports that solve $\min_{S \in P_i(\mathcal{D})} \mathcal{R}(S, \mathcal{D})$ for all $i \in \{0, 1, ..., s\}$.

# 4 Optimization Algorithms

In this section, we discuss how the top $R$ supports, $\hat{S}_1(\mathcal{D}), \cdots, \hat{S}_R(\mathcal{D})$, which are solutions to the problems in (5), can be obtained by solving a series of MIPs.

For clarity, we present the case of least squares objective. However, it is worth noting that a key benefit of our MIP approach is its generalizability across different loss functions. Specifically, the only property that we need to have for the loss function $\ell$ is that it is a convex function of $\beta$. We discuss any needed modifications for the case of hinge loss in Appendix C, and our method works with Huber or quantile loss as well, and the MIP algorithm would remain unchanged.

To obtain $\hat{S}_k(\mathcal{D})$ for $k \in [R]$ consider:

$$(\boldsymbol{z}^{(k)}, \boldsymbol{\beta}^{(k)}, \boldsymbol{\theta}^{(k)}) \in \underset{\boldsymbol{z}, \boldsymbol{\beta}, \boldsymbol{\theta}}{\arg\min} \quad \|\boldsymbol{y} - \boldsymbol{X}\boldsymbol{\beta}\|_2^2 \tag{9}$$

$$\text{s.t.} \quad \boldsymbol{\beta}, \boldsymbol{\theta} \in \mathbb{R}^p, \boldsymbol{z} \in \{0,1\}^p, \ \boldsymbol{\theta} \geq 0, \ \sum_{i=1}^p z_i = s, \ \sum_{i=1}^p \theta_i \leq r^2,$$

$$\beta_i^2 \leq \theta_i z_i \ \forall i \in [p], \ \sum_{i \in \hat{S}_j(\mathcal{D})} z_i \leq s - \frac{1}{2}, \ j \in [k-1].$$

In the following proposition, we show that Problems (5) and (9) are equivalent.

**Proposition 1.** *For $k \geq 1$, $\{i : z_i^{(k)} \neq 0\} = \hat{S}_k(\mathcal{D})$.*

Problem (9) can be solved to optimality using off-the-shelf solvers like Gurobi for moderately-sized datasets. In order to run our DP methods in even higher dimensions, where $p = 10{,}000$, we present a more tailored algorithm for solving the MIPs in the remainder of this section. This adds to the line of work on developing specialized discrete optimization algorithms for solving sparse regression problems and relatives in the non-private setting—see for eg [18, 17, 19, 4, 1].

We first add a ridge penalty term to the objective, which makes it strongly convex and a function of $\boldsymbol{z}$. To obtain $\hat{S}_k(\mathcal{D})$ for $k \in [R]$, we define

$$c(\boldsymbol{z}) = \min_{\|\boldsymbol{\beta}\|_2^2 \leq r^2} \frac{1}{2n} \|\boldsymbol{y} - \boldsymbol{X}\boldsymbol{\beta}\|_2^2 + \frac{\lambda}{2n} \sum_{i=1}^p \frac{\beta_i^2}{z_i}$$

and we seek to solve

$$\min_{\boldsymbol{z}} \ c(\boldsymbol{z}) \tag{10}$$

$$\text{subject to} \ \boldsymbol{z} \in \{0,1\}^p, \ \sum_{i=1}^p z_i = s, \ \sum_{i \in \hat{S}_j(\mathcal{D})} z_i \leq s - \frac{1}{2} \ \forall j \in [k-1].$$

For any $\boldsymbol{z} \in \{0,1\}^p$, let $\hat{z}_i = z_i$ if $z_i = 1$ and $\hat{z}_i = z_i + \text{Unif}[a,b]$ if $z_i = 0$, where $a > 0$ and $b < 1$. Let

$$\hat{\boldsymbol{\beta}} \in \arg \min_{\|\boldsymbol{\beta}\|_2^2 \leq r^2} \frac{1}{2n} \|\boldsymbol{y} - \boldsymbol{X}\boldsymbol{\beta}\|_2^2 + \frac{\lambda}{2n} \sum_{i=1}^p \frac{\beta_i^2}{\hat{z}_i}.$$

By Danskin's theorem [3], we then have

$$(\nabla c(\hat{\boldsymbol{z}}))_i = -\frac{\lambda}{2n} \frac{(\hat{\beta}_i)^2}{\hat{z}_i^2}.$$

Taking $\hat{\boldsymbol{z}}_0, \hat{\boldsymbol{z}}_1, ..., \hat{\boldsymbol{z}}_t \in (0,1]^p$, we have by convexity of $c$ that for all $\boldsymbol{x} \in (0,1]^p$ and for all $k \in \{0, ..., t\}$,

$$c(\boldsymbol{x}) \geq c(\hat{\boldsymbol{z}}_k) + \nabla c(\hat{\boldsymbol{z}}_k)^T (\boldsymbol{x} - \hat{\boldsymbol{z}}_k).$$

So then the map

$$c_t(\boldsymbol{x}) = \max\{c(\hat{\boldsymbol{z}}_0) + \nabla c(\hat{\boldsymbol{z}}_0)^T(\boldsymbol{x} - \hat{\boldsymbol{z}}_0), ..., c(\hat{\boldsymbol{z}}_t) + \nabla c(\hat{\boldsymbol{z}}_t)^T(\boldsymbol{x} - \hat{\boldsymbol{z}}_t)\}$$

is a lower bound on the map $c$. We can now present our outer approximation algorithm for solving Problem 10, based on [8, 4].

---

**Algorithm 2** Outer approximation for $\hat{S}_k(\mathcal{D})$

---

1: **procedure** $\mathcal{A}(\mathcal{D}, \lambda, r, s, a, b, \text{tol})$
2:     Initialize $z_0 \in \{0, 1\}^p$ s.t. $\sum_{i=1}^p z_i \le s$, $\eta_0 \leftarrow 0$, $t \leftarrow 0$
3:     $\hat{z}_0 \leftarrow \texttt{add\_noise}(z_0)$
4:     **while** $\frac{|\eta_t - c(z_t)|}{c(z_t)} > \text{tol}$ **do**

5:         $z_{t+1}, \eta_{t+1} \leftarrow \arg\min_{z \in \{0,1\}^p, \eta}
\begin{cases}
\eta & \\
\text{s.t. } \sum_{i=1}^p z_i \le s, \ \sum_{i \in \hat{S}_j(\mathcal{D})} z_i \le s - \frac{1}{2} \quad \forall j \in [k-1], & \\
\eta \ge c(\hat{z}_k) + \nabla c(\hat{z}_k)^T(z - \hat{z}_k) \quad \forall k \in [t] &
\end{cases}$

6:         $\hat{z}_{t+1} \leftarrow \texttt{add\_noise}(z_{t+1})$
7:         $t \leftarrow t + 1$
    **return** $z_t$

---

Intuitively, this approach seeks to solve Problem 10 by constructing a sequence of MIP approximations based on cutting planes. At each iteration, the cutting plane $\eta \ge c(\hat{z}_k) + \nabla c(\hat{z}_k)^T(z - \hat{z}_k)$ is added, cutting off $\hat{z}_t$, the current noisy version of the binary solution $z_t$, unless $z_t$ happened to be optimal as defined by our stopping criterion, which is $\frac{|\eta_t - c(z_t)|}{c(z_t)} \le \text{tol}$. As the algorithm progresses, the outer approximation function $c_t(z) = \max_{i \in [t]} c(\hat{z}_i) + \nabla c(\hat{z}_i)^\top(z - \hat{z}_i)$ becomes an increasibly better approximation to the loss function $c$, making our lower bound $\eta_t$ converge to the upper bound obtained by evaluating $c(z_t)$. Please refer to Appendix B for additional algorithmic details for the setting of BSS, and refer to Appendix C for the necessary modifications in the case of hinge loss.

**Remark 2.** The approach for obtaining $\tilde{S}_k(\mathcal{D})$, where $k \in [s]$, is very analogous to Algorithm 2, except instead of having the constraints $\sum_{i \in \hat{S}_j(\mathcal{D})} z_i \le s - \frac{1}{2} \ \forall j \in [l-1]$, where $l \in [R]$, we have the constraint $\sum_{i \in \tilde{S}_0(\mathcal{D})} z_i \le s - (k-1) - \frac{1}{2}$.

## 5 Numerical Experiments

In our experiments, we draw the data points as $y_i = x_i^T \beta^* + \epsilon_i$ for $i \in [n]$, where $x_1, \cdots, x_n \overset{\text{iid}}{\sim} \mathcal{N}(\mathbf{0}, \mathbf{\Sigma}) \in \mathbb{R}^p$ and the independent noise follows $\epsilon \sim \mathcal{N}(\mathbf{0}, \sigma^2 \mathbf{I}_n)$ where $\mathbf{I}_n$ is the identity matrix of size $n$. Moreover, for $i, j \in [p]$, we set $\Sigma_{i,j} = \rho^{|i-j|}$ and set nonzero coordinates of $\beta^*$ to take value $1/\sqrt{s}$ at indices $\{1, 3, \cdots, 2s-1\}$. We define the Signal to Noise Ratio as $\text{SNR} = \|X\beta^*\|_2^2 / \|\epsilon\|_2^2$. In Algorithm 1, we set $R = 2 + (p-s)s$, $b_x = b_y = 0.5$, $r = 1.1$ and $T = \infty$ for all our experiments in this paper. In Algorithm 2, we set $a = 0.001, b = 0.005, r = 1.1$ and $\text{tol} = 0.005$, and consider various values of the other parameters.

In Figure 1a, we plot the average proportion of draws from 10 independent trials that recovered the right support for our top-$R$ and mistakes methods using least squares as our objective for $p = 10,000$. We compare with the MCMC algorithm from [29] and the Samp-Agg algorithm in [23], wherein we use Lasso for the $\mathcal{A}_{\text{supp}}$ subroutine. In Figure 1b, we show the analogous results for hinge loss, comparing with Lasso Samp-Agg. More experimental results with varying values of SNR, $p, s, \rho$, and $\epsilon$ for least squares and hinge loss, as well as results on prediction accuracy, utility loss, and ablation studies are provided in Appendix D. For each trial, we drew 50 times from the distribution corresponding to each algorithm and gathered the proportion of correct supports. For MCMC, we similarly used 50 independent Markov chains from random initialization and gathered the proportion of correct supports after a number of iterations that was chosen to make the runtime comparable to our methods.

We have that in all settings, both of our methods outperform other algorithms for large enough $n$. The proportion of draws that recover the right support increases with $n$, since a larger sample size reduces the threshold required for the identifiability margin $\tau$ (discussed in Section 3) to have enough separation between the true support and other supports. Furthermore, performance improves at much

larger values of $n$ when $p$ or $s$ is greater or $\epsilon$ is lower, since the lower bound on $\tau$ is harder to satisfy in those settings. Moreover, we observe that, keeping $s, n, p$, and $\epsilon$ fixed, smaller values of SNR and larger values of $\rho$ make recovery harder, as $\tau$ decreases with lower signal strength or more correlation between features. Furthermore, the mistakes method numerically outperforms top-R, aligning with Section 3, which shows it succeeds under a milder identifiability condition.

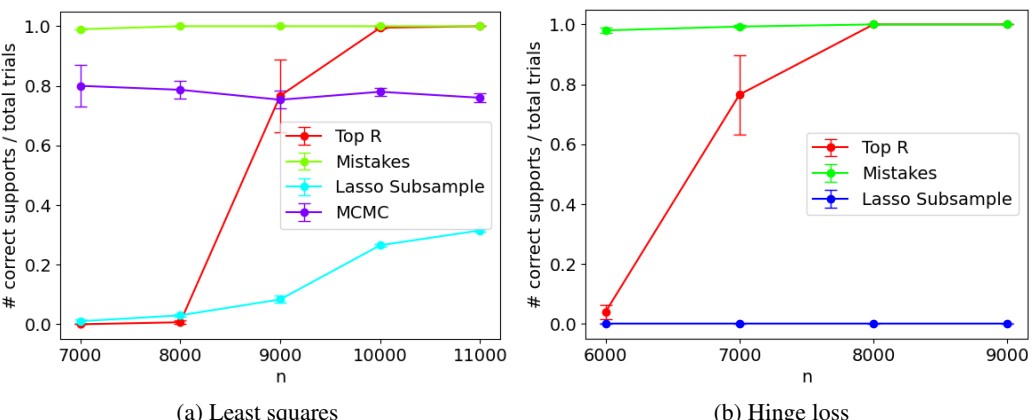

(a) Least squares            (b) Hinge loss

Figure 1: Simulations for $p = 10{,}000$, $s = 5$, SNR=5, $\rho = 0.1$, and $\epsilon = 1$ for least squares and hinge loss. The penalty parameter $\lambda$ in Algorithm 2 was set to 600 and 170 for figures 1a and 1b, respectively, and the number of MCMC iterations was set to $100{,}000$ for 1a. On the $x$-axis, we vary the value of $n$ and plot the average proportion of draws across 10 independent trials that recovered the right support for each corresponding algorithm. Error bars denote the mean standard error.

**Computational resources and license information:** All experiments were conducted on a computing cluster using 20 cores and 64 GB RAM. The Gurobi Optimizer is used under the Gurobi End User License Agreement. CVXPY is distributed under the Apache License, Version 2.0. ABESS package is distributed under GNU General Public License, Version 3.

# 6 Conclusion

In this paper, we introduced two scalable pure DP estimators for variable selection in sparse high-dimensional settings. While we provide utility guarantees specific to the BSS setting, we demonstrate how our methods can be applied more broadly, yielding favorable support recovery in the additional setting of sparse classification with hinge loss. Our contributions enhance privacy-preserving practices, enabling safer use of sensitive datasets in critical areas such as medicine, public health, finance, and personalized recommendation systems.

One limitation of our work is that Theorem 2 requires an additional assumption that holds with high probability for $\tau$ large enough, and it remains an open question whether a lighter assumption can be made to yield privacy guarantees for the mistakes method. Furthermore, our theoretical support recovery results yield sufficient conditions for support recovery, but an interesting direction of research may be to find necessary conditions as well, to see if our bounds on $\tau$ are tight.

# 7 Acknowledgements

We thank the authors of [29] for sharing their code with us. This research is supported in part by grants from the Office of Naval Research (N000142512504, N000142212665). A shorter workshop version of the paper appeared in [2]. The research started when Kayhan Behdin was a PhD student at MIT.

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

# A  Proofs of Main Results

## A.1  Lemma A.1

**Lemma A.1.** *Suppose Assumption 1 holds, with*

$$\mathcal{R}(S, \mathcal{D}) = \min_{\boldsymbol{\beta} \in \mathbb{R}^{|S|}} \|\boldsymbol{y} - \boldsymbol{X}_S \boldsymbol{\beta}\|_2^2 \text{ s.t. } \|\boldsymbol{\beta}\|_2^2 \le r^2. \tag{A.1}$$

*Then,*

$$\Delta \le 2b_y^2 + 2b_x^2 r^2 s.$$

*Proof.* Suppose $\mathcal{D}, \mathcal{D}'$ are two neighboring datasets. Fix a support $S \in \mathcal{O}$ and suppose

$$\hat{\boldsymbol{\beta}} \in \arg\min_{\boldsymbol{\beta} \in \mathbb{R}^s} \|\boldsymbol{y}' - \boldsymbol{X}_S' \boldsymbol{\beta}\|_2^2 \text{ s.t. } \|\boldsymbol{\beta}\|_2^2 \le r^2.$$

Then,

$$\mathcal{R}(S, \mathcal{D}) - \mathcal{R}(S, \mathcal{D}') \le \|\boldsymbol{y} - \boldsymbol{X}_S \hat{\boldsymbol{\beta}}\|_2^2 - \|\boldsymbol{y}' - \boldsymbol{X}_S' \hat{\boldsymbol{\beta}}\|_2^2.$$

Let us assume without loss of generality that $\mathcal{D}, \mathcal{D}'$ differ in the $n$-th observations. Hence, we have that

$$
\begin{aligned}
\|\boldsymbol{y} - \boldsymbol{X}_S \hat{\boldsymbol{\beta}}\|_2^2 - \|\boldsymbol{y}' - \boldsymbol{X}_S' \hat{\boldsymbol{\beta}}\|_2^2 &= \sum_{i=1}^{n-1} [(y_i - (\boldsymbol{x}_i)_S^T \hat{\boldsymbol{\beta}})^2 - (y_i - (\boldsymbol{x}_i)_S^T \hat{\boldsymbol{\beta}})^2] \\
&\quad + (y_n - (\boldsymbol{x}_n)_S^T \hat{\boldsymbol{\beta}})^2 - (y_n' - (\boldsymbol{x}_n)_S'^T \hat{\boldsymbol{\beta}})^2 \\
&\le (y_n - (\boldsymbol{x}_n)_S^T \hat{\boldsymbol{\beta}})^2 \\
&\le 2y_n^2 + 2((\boldsymbol{x}_n)_S^T \hat{\boldsymbol{\beta}})^2 \\
&\le 2b_y^2 + 2b_x^2 r^2 s
\end{aligned}
$$

where the last step uses the Cauchy-Schwartz inequality and the fact that $|S| = s$. $\qquad\square$

## A.2  Proof of Theorem 1

First, we prove some technical results that will be used in the proof of Theorem 1. Define

$$\hat{\mathcal{R}}(S, \mathcal{D}) = \begin{cases} \mathcal{R}(S, \mathcal{D}) & \text{if } S \in \{\hat{S}_1(\mathcal{D}), \cdots, \hat{S}_R(\mathcal{D})\} \\ \mathcal{R}(\hat{S}_R(\mathcal{D}), \mathcal{D}) & \text{otherwise.} \end{cases} \tag{A.2}$$

**Lemma A.2.** *Let $\Delta$ to be taken as in (2). Then,*

$$\max_{\substack{k \ge 1}} \max_{\substack{\mathcal{D}, \mathcal{D}' \in \mathcal{Z}^n \\ \mathcal{D}, \mathcal{D}' \text{ are neighbors}}} \mathcal{R}(\hat{S}_k(\mathcal{D}), \mathcal{D}) - \mathcal{R}(\hat{S}_k(\mathcal{D}'), \mathcal{D}') \le \Delta.$$

*Proof.* Fix $k \ge 1$ and let us consider the following cases:
**Case 1:** $\mathcal{R}(\hat{S}_k(\mathcal{D}), \mathcal{D}') \le \mathcal{R}(\hat{S}_k(\mathcal{D}'), \mathcal{D}')$. Then, by the definition of $\Delta$,

$$
\begin{aligned}
\mathcal{R}(\hat{S}_k(\mathcal{D}), \mathcal{D}) - \mathcal{R}(\hat{S}_k(\mathcal{D}'), \mathcal{D}') &\le \mathcal{R}(\hat{S}_k(\mathcal{D}), \mathcal{D}') - \mathcal{R}(\hat{S}_k(\mathcal{D}'), \mathcal{D}') + \Delta \\
&\le \Delta.
\end{aligned} \tag{A.3}
$$

**Case 2:** $\mathcal{R}(\hat{S}_k(\mathcal{D}), \mathcal{D}) \le \mathcal{R}(\hat{S}_k(\mathcal{D}'), \mathcal{D})$. Then,

$$
\begin{aligned}
\mathcal{R}(\hat{S}_k(\mathcal{D}), \mathcal{D}) - \mathcal{R}(\hat{S}_k(\mathcal{D}'), \mathcal{D}') &\le \mathcal{R}(\hat{S}_k(\mathcal{D}), \mathcal{D}) - \mathcal{R}(\hat{S}_k(\mathcal{D}'), \mathcal{D}) + \Delta \\
&\le \Delta.
\end{aligned} \tag{A.4}
$$

**Case 3:** $\mathcal{R}(\hat{S}_k(\mathcal{D}), \mathcal{D}) > \mathcal{R}(\hat{S}_k(\mathcal{D}'), \mathcal{D})$ and $\mathcal{R}(\hat{S}_k(\mathcal{D}), \mathcal{D}') > \mathcal{R}(\hat{S}_k(\mathcal{D}'), \mathcal{D}')$. Trivially, in this case we must have $\binom{p}{s} > k \ge 2$. Then, there must exist $S_0 \subseteq [p], |S_0| = s$ such that

$$\mathcal{R}(S_0, \mathcal{D}) \ge \mathcal{R}(\hat{S}_k(\mathcal{D}), \mathcal{D}), \mathcal{R}(S_0, \mathcal{D}') \le \mathcal{R}(\hat{S}_k(\mathcal{D}'), \mathcal{D}').$$

To this end, define

$$\mathbb{S}_1 = \{\hat{S}_1(\mathcal{D}), \cdots, \hat{S}_{k-1}(\mathcal{D})\}, \ \mathbb{S}_2 = \{\hat{S}_{k+1}(\mathcal{D}), \cdots\}, \ \mathbb{S}' = \{\hat{S}_1(\mathcal{D}'), \cdots, \hat{S}_{k-1}(\mathcal{D}')\}.$$

As $\mathcal{R}(\hat{S}_k(\mathcal{D}), \mathcal{D}') > \mathcal{R}(\hat{S}_k(\mathcal{D}'), \mathcal{D}')$, we have that $\hat{S}_k(\mathcal{D}) \notin \mathbb{S}'$ so $\mathbb{S}' \subset \mathbb{S}_1 \cup \mathbb{S}_2$. On the other hand, as $\mathcal{R}(\hat{S}_k(\mathcal{D}), \mathcal{D}) > \mathcal{R}(\hat{S}_k(\mathcal{D}'), \mathcal{D})$, $\hat{S}_k(\mathcal{D}') \in \mathbb{S}_1$ and as $\hat{S}_k(\mathcal{D}') \notin \mathbb{S}'$, we have $|\mathbb{S}' \cap \mathbb{S}_1| \leq k - 2$. As $|\mathbb{S}'| = k - 1$, we must have $|\mathbb{S}' \cap \mathbb{S}_2| \geq 1$ which proves the existence of $S_0$. Next, note that

$$\mathcal{R}(\hat{S}_k(\mathcal{D}), \mathcal{D}) - \mathcal{R}(\hat{S}_k(\mathcal{D}'), \mathcal{D}') \leq \mathcal{R}(S_0, \mathcal{D}) - \mathcal{R}(S_0, \mathcal{D}') \leq \Delta. \tag{A.5}$$

$\square$

**Lemma A.3.** *Let $\Delta$ to be taken as in (2). Then,*

$$\max_{\substack{S \subseteq [p] \\ |S| = s}} \max_{\substack{\mathcal{D}, \mathcal{D}' \in \mathcal{Z}^n \\ \mathcal{D}, \mathcal{D}' \text{ are neighbors}}} \hat{\mathcal{R}}(S, \mathcal{D}) - \hat{\mathcal{R}}(S, \mathcal{D}') \leq \Delta.$$

*Proof.* Suppose $S = \hat{S}_{k_1}(\mathcal{D}) = \hat{S}_{k_2}(\mathcal{D}')$. Let us consider the following cases:

**Case 1:** $k_1, k_2 \geq R$: Then, we have $\hat{\mathcal{R}}(S, \mathcal{D}) = \mathcal{R}(\hat{S}_R(\mathcal{D}), \mathcal{D})$ and $\hat{\mathcal{R}}(S, \mathcal{D}') = \mathcal{R}(\hat{S}_R(\mathcal{D}'), \mathcal{D}')$. Therefore,

$$\hat{\mathcal{R}}(S, \mathcal{D}) - \hat{\mathcal{R}}(S, \mathcal{D}') = \mathcal{R}(\hat{S}_R(\mathcal{D}), \mathcal{D}) - \mathcal{R}(\hat{S}_R(\mathcal{D}'), \mathcal{D}') \leq \Delta \tag{A.6}$$

by Lemma A.2.

**Case 2:** $k_1 < R, k_2 \geq R$: Then, we have $\hat{\mathcal{R}}(S, \mathcal{D}) = \mathcal{R}(S, \mathcal{D}) \leq \mathcal{R}(\hat{S}_R(\mathcal{D}), \mathcal{D})$ and $\hat{\mathcal{R}}(S, \mathcal{D}') = \mathcal{R}(\hat{S}_R(\mathcal{D}'), \mathcal{D}')$. Then,

$$\hat{\mathcal{R}}(S, \mathcal{D}) - \hat{\mathcal{R}}(S, \mathcal{D}') \leq \mathcal{R}(\hat{S}_R(\mathcal{D}), \mathcal{D}) - \mathcal{R}(\hat{S}_R(\mathcal{D}'), \mathcal{D}') \leq \Delta. \tag{A.7}$$

**Case 3:** $k_1 \geq R, k_2 < R$: Then, we have $\hat{\mathcal{R}}(S, \mathcal{D}) = \mathcal{R}(\hat{S}_R(\mathcal{D}), \mathcal{D}) \leq \mathcal{R}(S, \mathcal{D})$ and $\hat{\mathcal{R}}(S, \mathcal{D}') = \mathcal{R}(S, \mathcal{D}')$. Then,

$$\hat{\mathcal{R}}(S, \mathcal{D}) - \hat{\mathcal{R}}(S, \mathcal{D}') \leq \mathcal{R}(S, \mathcal{D}) - \mathcal{R}(S, \mathcal{D}') \leq \Delta. \tag{A.8}$$

**Case 4:** $k_1, k_2 < R$: Then, we have $\hat{\mathcal{R}}(S, \mathcal{D}) = \mathcal{R}(S, \mathcal{D})$ and $\hat{\mathcal{R}}(S, \mathcal{D}') = \mathcal{R}(S, \mathcal{D}')$. The result follows. $\square$

**Lemma A.4.** *Suppose $\hat{\mathcal{M}}$ is as defined in Algorithm 1, and $\hat{\mathcal{A}}_E$ is an exponential mechanism with the objective $\hat{\mathcal{R}}$,*

$$\mathbb{P}(\hat{\mathcal{A}}_E(\mathcal{D}) = S) \propto \exp\left(-\frac{\varepsilon \hat{\mathcal{R}}(S, \mathcal{D})}{2\Delta}\right), \ \forall S \in \mathcal{O}. \tag{A.9}$$

*Then, for $S \in \mathcal{O}$,*

$$(1 - q^T)\mathbb{P}(\hat{\mathcal{A}}_E(\mathcal{D}) = S) \leq \mathbb{P}(\hat{\mathcal{M}}(\mathcal{D}) = S) \leq \mathbb{P}(\hat{\mathcal{A}}_E(\mathcal{D}) = S) + q^T \tag{A.10}$$

*where*

$$q = \frac{R}{\binom{p}{s}}.$$

*Proof.* Fix $S \in \mathcal{O}$ and suppose $S = \hat{S}_k(\mathcal{D})$. Moreover, let $\mathbb{S}_R = \{\hat{S}_k(\mathcal{D}) : k \leq R\}$. Consider the following cases:

**Case 1:** $k \leq R$: Then, based on Algorithm 1,

$$\mathbb{P}(\hat{\mathcal{M}}(\mathcal{D}) = S) = \mathbb{P}\left(\{a(\mathcal{D}) = k\} \cup \{a(\mathcal{D}) = R + 1, \mathcal{M}_0(\mathcal{D}) = S\}\right).$$

Therefore,

$$\mathbb{P}(a(\mathcal{D}) = k) \leq \mathbb{P}(\hat{\mathcal{M}}(\mathcal{D}) = S) = \mathbb{P}\left(\{a(\mathcal{D}) = k\} \cup \{a(\mathcal{D}) = R + 1, \mathcal{M}_0(\mathcal{D}) = S\}\right)$$
$$\overset{(a)}{\leq} \mathbb{P}(a(\mathcal{D}) = k) + \mathbb{P}(\mathcal{M}_0(\mathcal{D}) \in \mathbb{S}_R)$$
$$\overset{(b)}{\leq} \mathbb{P}(a(\mathcal{D}) = k) + q^T \tag{A.11}$$

where $(a)$ is true as $S \in \mathbb{S}_R$, and $(b)$ is true as $\mathcal{M}_0$ return a support in $\mathbb{S}_R$ if it selects some support from $\mathbb{S}_R$ for all $T$ iterations, showing $\mathbb{P}(\mathcal{M}_0(\mathcal{D}) \in \mathbb{S}_R) = q^T$. Note that for $k \leq R$, $\mathbb{P}(a(\mathcal{D}) = k) = \mathbb{P}(\hat{\mathcal{A}}_E(\mathcal{D}) = S)$, therefore,

$$\mathbb{P}(\hat{\mathcal{A}}_E(\mathcal{D}) = S) \leq \mathbb{P}(\hat{\mathcal{M}}(\mathcal{D}) = S) \leq \mathbb{P}(\hat{\mathcal{A}}_E(\mathcal{D}) = S) + q^T. \tag{A.12}$$

**Case 2:** $k > R$**:** Then, from (6),

$$
\begin{aligned}
\mathbb{P}(a(\mathcal{D}) = R+1) &= \frac{\left(\binom{p}{s} - R\right) \exp\left(-\frac{\varepsilon}{2\Delta}\mathcal{R}(\hat{S}_R(\mathcal{D}), \mathcal{D})\right)}{\sum_{k=1}^{R} \exp\left(-\frac{\varepsilon}{2\Delta}\mathcal{R}(\hat{S}_k(\mathcal{D}), \mathcal{D})\right) + \left(\binom{p}{s} - R\right) \exp\left(-\frac{\varepsilon}{2\Delta}\mathcal{R}(\hat{S}_R(\mathcal{D}), \mathcal{D})\right)} \\
&= \frac{\left(\binom{p}{s} - R\right) \exp\left(-\frac{\varepsilon}{2\Delta}\hat{\mathcal{R}}(S, \mathcal{D})\right)}{\sum_{k=1}^{R} \exp\left(-\frac{\varepsilon}{2\Delta}\hat{\mathcal{R}}(\hat{S}_k(\mathcal{D}), \mathcal{D})\right) + \sum_{k \geq R+1} \exp\left(-\frac{\varepsilon}{2\Delta}\hat{\mathcal{R}}(\hat{S}_k(\mathcal{D}), \mathcal{D})\right)} \\
&= \left(\binom{p}{s} - R\right)\mathbb{P}(\hat{\mathcal{A}}_E(\mathcal{D}) = S). \tag{A.13}
\end{aligned}
$$

Hence, one can write

$$
\begin{aligned}
\mathbb{P}(\hat{\mathcal{M}}(\mathcal{D}) = S) &= \mathbb{P}\left(\{a(\mathcal{D}) = R+1\} \cap \{\mathcal{M}_0(\mathcal{D}) = S\}\right) \\
&\overset{(a)}{=} \mathbb{P}(a(\mathcal{D}) = R+1)\left(\sum_{i=1}^{T} \frac{q^{i-1}}{\binom{p}{s}}\right) \\
&= \frac{1}{\binom{p}{s}}\left[\binom{p}{s} - R\right]\mathbb{P}(\hat{\mathcal{A}}_E(\mathcal{D}) = S)\frac{1 - q^T}{1 - q} \\
&= (1 - q^T)\mathbb{P}(\hat{\mathcal{A}}_E(\mathcal{D}) = S) \tag{A.14}
\end{aligned}
$$

where $(a)$ is true as

$$\mathbb{P}(\mathcal{M}_0(\mathcal{D}) = S) = \sum_{i=1}^{T} \mathbb{P}(\mathcal{M}_0(\mathcal{D}) = S, \mathcal{M}_0 \text{ stops after } i \text{ iterations }) = \sum_{i=1}^{T} \frac{q^{i-1}}{\binom{p}{s}}.$$

The proof is complete by (A.12) and (A.14). $\qquad\square$

Next, let us prove an important intermediate result on Algorithm 1.

**Theorem A.1.** *Suppose* $T > 1$, $1 < R < \binom{p}{s}$. *The procedure* $\hat{\mathcal{M}}$ *in Algorithm 1 is* $(\varepsilon', \delta)$*-DP where*

$$\varepsilon' = \varepsilon - \log\left(1 - \left[R/\binom{p}{s}\right]^T\right), \quad \delta = \left[R/\binom{p}{s}\right]^T.$$

*Proof.* From Lemma A.3 and Lemma 1, we know that $\hat{\mathcal{A}}_E$ is an $(\varepsilon, 0)$-DP procedure. Suppose $\mathcal{D}, \mathcal{D}'$ are neighboring datasets. Then, from Lemma A.4,

$$
\begin{aligned}
\mathbb{P}(\hat{\mathcal{M}}(\mathcal{D}) = S) &\leq \mathbb{P}(\hat{\mathcal{A}}_E(\mathcal{D}) = S) + q^T \\
&\leq e^\varepsilon \mathbb{P}(\hat{\mathcal{A}}_E(\mathcal{D}') = S) + q^T \\
&\leq \frac{1}{1 - q^T} e^\varepsilon \mathbb{P}(\hat{\mathcal{M}}(\mathcal{D}') = S) + q^T \tag{A.15}
\end{aligned}
$$

where the first and last inequality use Lemma A.4. $\qquad\square$

***Proof of Theorem 1.*** Note that by definition, for $S \in \mathcal{O}$, we have $0 \leq \mathcal{R}(S, \mathcal{D}) \leq \|\boldsymbol{y}\|_2^2 \leq nb_y^2$. Then,

$$
\begin{aligned}
\mathbb{P}(\hat{\mathcal{A}}_E(\mathcal{D}) = S) &= \frac{\exp\left(-\frac{\varepsilon}{2\Delta}\hat{\mathcal{R}}(S, \mathcal{D})\right)}{\sum_{k=1}^{R} \exp\left(-\frac{\varepsilon}{2\Delta}\hat{\mathcal{R}}(\hat{S}_k(\mathcal{D}), \mathcal{D})\right) + \sum_{k \geq R+1} \exp\left(-\frac{\varepsilon}{2\Delta}\hat{\mathcal{R}}(\hat{S}_k(\mathcal{D}), \mathcal{D})\right)} \\
&\geq \frac{\exp(-n\varepsilon b_y^2/(2\Delta))}{\binom{p}{s}} := \delta_0. \tag{A.16}
\end{aligned}
$$

Then, from (A.15),

$$\mathbb{P}(\hat{\mathcal{M}}(\mathcal{D}) = S) \leq e^\varepsilon \mathbb{P}(\hat{\mathcal{A}}_E(\mathcal{D}') = S) + q^T$$

$$= \left(e^\varepsilon + \frac{q^T}{\delta_0}\right) \mathbb{P}(\hat{\mathcal{A}}_E(\mathcal{D}') = S) - \frac{q^T}{\delta_0}\mathbb{P}(\hat{\mathcal{A}}_E(\mathcal{D}') = S) + q^T$$

$$\overset{(a)}{\leq} \left(e^\varepsilon + \frac{q^T}{\delta_0}\right) \mathbb{P}(\hat{\mathcal{A}}_E(\mathcal{D}') = S)$$

$$\overset{(b)}{\leq} \frac{1}{1 - q^T}\left(e^\varepsilon + \frac{q^T}{\delta_0}\right) \mathbb{P}(\hat{\mathcal{M}}(\mathcal{D}') = S) \qquad (A.17)$$

where $(a)$ is by (A.16) and $(b)$ is by Lemma A.4. $\qquad\square$

## A.3  Effect of choice of $R$ on support recovery for top-$R$ method

In this section, we formalize the intuition discussed in Section 2.1.1 regarding the impact that the choice of $R$ has on the top-$R$ recovering the optimal BSS support $\hat{S}_1(\mathcal{D})$. We show that, as $R$ increases, the probability of top-$R$ outputting $\hat{S}_1(\mathcal{D})$ can only improve.

**Lemma A.5.** *Take $T = \infty$ in Algorithm 1. Denote $\hat{\mathcal{M}}_1$ and $\hat{\mathcal{M}}_2$ as two instances of the top-$R$ method, using $R_1$ and $R_2$ enumerated supports, respectively, where $R_1 < R_2$. Then, we have that*

$$\mathbb{P}[\hat{\mathcal{M}}_1(\mathcal{D}) = \hat{S}_1(\mathcal{D})] \leq \mathbb{P}[\hat{\mathcal{M}}_2(\mathcal{D}) = \hat{S}_1(\mathcal{D})].$$

*Proof.* By Lemma A.4, we have that, when $T = \infty$,

$$\mathbb{P}[\hat{\mathcal{A}}_E(\mathcal{D}) = \hat{S}_1(\mathcal{D})] = \mathbb{P}[\hat{\mathcal{M}}(\mathcal{D}) = \hat{S}_1(\mathcal{D})]$$

Let $\hat{\mathcal{A}}_{E_1}$ and $\hat{\mathcal{A}}_{E_2}$ denote the exponential mechanisms with the objective $\hat{\mathcal{R}}$ using $R_1$ and $R_2$ enumerated supports, respectively. It then suffices to show that

$$\mathbb{P}[\hat{\mathcal{A}}_{E_1}(\mathcal{D}) = \hat{S}_1(\mathcal{D})] \leq \mathbb{P}[\hat{\mathcal{A}}_{E_2}(\mathcal{D}) = \hat{S}_1(\mathcal{D})].$$

Define $G_i := \mathcal{R}(\hat{S}_i(\mathcal{D}), \mathcal{D}) - \mathcal{R}(\hat{S}_1(\mathcal{D}), \mathcal{D})$. Then, note that

$$\mathbb{P}(\hat{\mathcal{A}}_{E_1}(\mathcal{D}) = \hat{S}_1(\mathcal{D})) = \frac{1}{1 + \sum_{i=2}^{R_1}\exp(-\epsilon G_i/(2\Delta)) + (\binom{p}{s} - R_1)\exp(-\epsilon G_{R_1}/(2\Delta))}$$

and

$$\mathbb{P}(\hat{\mathcal{A}}_{E_2}(\mathcal{D}) = \hat{S}_1(\mathcal{D})) = \frac{1}{1 + \sum_{i=2}^{R_2}\exp(-\epsilon G_i/(2\Delta)) + (\binom{p}{s} - R_2)\exp(-\epsilon G_{R_2}/(2\Delta))}.$$

We then have that

$$\sum_{i=R_1+1}^{R_2}\exp\left(-\frac{\epsilon G_i}{2\Delta}\right) \leq (R_2 - R_1)\exp\left(-\frac{\epsilon G_{R_1}}{2\Delta}\right)$$

and

$$(\binom{p}{s} - R_2)\exp\left(-\frac{\epsilon G_{R_2}}{2\Delta}\right) \leq (\binom{p}{s} - R_2)\exp\left(-\frac{\epsilon G_{R_1}}{2\Delta}\right)$$

since the supports are sorted in increasing objective value. The result follows. $\qquad\square$

## A.4  Proof of Theorem 2

*Proof.* First, we claim that if $\mathcal{R}(\hat{S}_2(\mathcal{D}), \mathcal{D}) - \mathcal{R}(\hat{S}_1(\mathcal{D}), \mathcal{D}) > 2\Delta$, then $\hat{S}_1(\mathcal{D}) = \hat{S}_1(\mathcal{D}')$ for neighboring datasets $\mathcal{D}, \mathcal{D}'$.

Suppose, by way of contradiction, that $\hat{S}_1(\mathcal{D}) \neq \hat{S}_1(\mathcal{D}')$. Then we have that $\mathcal{R}(\hat{S}_1(\mathcal{D}'), \mathcal{D}') \leq \mathcal{R}(\hat{S}_1(\mathcal{D}), \mathcal{D}')$ and $\mathcal{R}(\hat{S}_1(\mathcal{D}'), \mathcal{D}) \geq \mathcal{R}(\hat{S}_2(\mathcal{D}), \mathcal{D})$. By definition of $\Delta$, we have that

$\mathcal{R}(\hat{S}_1(\mathcal{D}), \mathcal{D}') \leq \mathcal{R}(\hat{S}_1(\mathcal{D}), \mathcal{D}) + \Delta$ and $\mathcal{R}(\hat{S}_1(\mathcal{D}'), \mathcal{D}) - \Delta \leq \mathcal{R}(\hat{S}_1(\mathcal{D}'), \mathcal{D}')$. Combining we have

$$\mathcal{R}(\hat{S}_1(\mathcal{D}), \mathcal{D}) + \Delta \geq \mathcal{R}(\hat{S}_1(\mathcal{D}), \mathcal{D}') \geq \mathcal{R}(\hat{S}_1(\mathcal{D}'), \mathcal{D}') \geq \mathcal{R}(\hat{S}_1(\mathcal{D}'), \mathcal{D}) - \Delta \geq \mathcal{R}(\hat{S}_2(\mathcal{D}), \mathcal{D}) - \Delta.$$

Thus $2\Delta \geq \mathcal{R}(\hat{S}_2(\mathcal{D}), \mathcal{D}) - \mathcal{R}(\hat{S}_1(\mathcal{D}), \mathcal{D})$. This is a contradiction with the assumption that $\mathcal{R}(\hat{S}_2(\mathcal{D}), \mathcal{D}) - \mathcal{R}(\hat{S}_1(\mathcal{D}), \mathcal{D}) > 2\Delta$.

Since $\hat{S}_1(\mathcal{D}) = \hat{S}_1(\mathcal{D}')$, i.e. $\tilde{S}_0(\mathcal{D}) = \tilde{S}_0(\mathcal{D}')$, we have that $P_i(\mathcal{D}) = P_i(\mathcal{D}')$ for all $i \in \{0, 1, ..., s\}$.

Suppose $S \in P_k(\mathcal{D}), P_k(\mathcal{D}')$. Then, by definition of $\tilde{\mathcal{R}}$, we have $\tilde{\mathcal{R}}(S, \mathcal{D}) = \mathcal{R}(\tilde{S}_k(\mathcal{D}), \mathcal{D})$ and $\tilde{\mathcal{R}}(S, \mathcal{D}') = \mathcal{R}(\tilde{S}_k(\mathcal{D}'), \mathcal{D}')$. We have that $\mathcal{R}(\tilde{S}_k(\mathcal{D}), \mathcal{D}) = \min_{S \in P_k(\mathcal{D})} \mathcal{R}(S, \mathcal{D})$. Since $\tilde{S}_k(\mathcal{D}') \in P_k(\mathcal{D})$, we have that $\mathcal{R}(\tilde{S}_k(\mathcal{D}), \mathcal{D}) \leq \mathcal{R}(\tilde{S}_k(\mathcal{D}'), \mathcal{D})$.

Then, by definition of $\Delta$, we have that

$$\tilde{\mathcal{R}}(S, \mathcal{D}) - \tilde{\mathcal{R}}(S, \mathcal{D}') = \mathcal{R}(\tilde{S}_k(\mathcal{D}), \mathcal{D}) - \mathcal{R}(\tilde{S}_k(\mathcal{D}'), \mathcal{D}') \leq \mathcal{R}(\tilde{S}_k(\mathcal{D}), \mathcal{D}) - \mathcal{R}(\tilde{S}_k(\mathcal{D}'), \mathcal{D}) + \Delta \leq \Delta.$$

Thus, we have that

$$\max_{\substack{S \subseteq [p] \\ |S| = s}} \max_{\substack{\mathcal{D}, \mathcal{D}' \in \mathcal{Z}^n \\ \mathcal{D}, \mathcal{D}' \text{ are neighbors}}} \tilde{\mathcal{R}}(S, \mathcal{D}) - \tilde{\mathcal{R}}(S, \mathcal{D}') \leq \Delta.$$

Since $\tilde{\mathcal{R}}$ has bounded global sensitivity $\Delta$, we have that, by Lemma 1, the mistakes method, which is the exponential mechanism with scoring function $\tilde{\mathcal{R}}$, is $(\epsilon, 0)$-differentially private. $\square$

## A.5 Sufficient conditions for privacy of mistakes method

Below, we present sufficient conditions under which $\mathcal{R}(\hat{S}_2(\mathcal{D}), \mathcal{D}) - \mathcal{R}(\hat{S}_1(\mathcal{D}), \mathcal{D}) > 2\Delta$ with high probability, which, by Theorem 2, implies that the mistakes method is $(\epsilon, 0)$-DP with high probability.

**Lemma A.6.** *Suppose that assumptions 1-4 hold. Set $r \geq (\frac{\kappa_+}{\kappa_-})M + 4\frac{\sigma b_x}{\kappa_-}$. Then, there exists a universal constant $C > 0$ such that, whenever*

$$\tau \geq C\sigma^2 \frac{s \log p}{n},$$

*we have that*

$$\mathbb{P}(\mathcal{R}(\hat{S}_2(\mathcal{D}), \mathcal{D}) - \mathcal{R}(\hat{S}_1(\mathcal{D}), \mathcal{D}) > 2\Delta) \geq 1 - 10sp^{-2}.$$

*Proof.* Following the proof of Theorem 2.1 from [13], we have that

$$\begin{aligned}
n^{-1}(\mathcal{R}_{ols}(S, \mathcal{D}) - \mathcal{R}_{ols}(S^*, \mathcal{D})) &= n^{-1}\{\boldsymbol{y}^\top(\boldsymbol{I} - \boldsymbol{P}_{X_S})\boldsymbol{y} - \boldsymbol{y}^\top(\boldsymbol{I} - \boldsymbol{P}_{X_{S^*}})\boldsymbol{y}\} \\
&= n^{-1}\{(\boldsymbol{X}_{S^*\backslash S}\boldsymbol{\beta}^*_{S^*\backslash S} + \boldsymbol{\epsilon})^\top(\boldsymbol{I} - \boldsymbol{P}_{X_S})(\boldsymbol{X}_{S^*\backslash S}\boldsymbol{\beta}^*_{S^*\backslash S} + \boldsymbol{\epsilon}) - \boldsymbol{\epsilon}^\top(\boldsymbol{I} - \boldsymbol{P}_{X_{S^*}})\boldsymbol{\epsilon}\} \\
&= \boldsymbol{\beta}^{*\top}_{S^*\backslash S}\hat{\boldsymbol{D}}(S)\boldsymbol{\beta}^*_{S^*\backslash S} + 2n^{-1}\boldsymbol{\epsilon}^\top(\boldsymbol{I} - \boldsymbol{P}_{X_S})\boldsymbol{X}_{S^*\backslash S}\boldsymbol{\beta}^*_{S^*\backslash S} - n^{-1}\boldsymbol{\epsilon}^\top(\boldsymbol{P}_{X_S} - \boldsymbol{P}_{X_{S^*}})\boldsymbol{\epsilon} \\
&= \frac{1}{2}\boldsymbol{\beta}^{*\top}_{S^*\backslash S}\hat{\boldsymbol{D}}(S)\boldsymbol{\beta}^*_{S^*\backslash S} + \frac{1}{4}\boldsymbol{\beta}^{*\top}_{S^*\backslash S}\hat{\boldsymbol{D}}(S)\boldsymbol{\beta}^*_{S^*\backslash S} + 2n^{-1}\boldsymbol{\epsilon}^\top(\boldsymbol{I} - \boldsymbol{P}_{X_S})\boldsymbol{X}_{S^*\backslash S}\boldsymbol{\beta}^*_{S^*\backslash S} \\
&\quad + \frac{1}{4}\boldsymbol{\beta}^{*\top}_{S^*\backslash S}\hat{\boldsymbol{D}}(S)\boldsymbol{\beta}^*_{S^*\backslash S} - n^{-1}\boldsymbol{\epsilon}^\top(\boldsymbol{P}_{X_S} - \boldsymbol{P}_{X_{S^*}})\boldsymbol{\epsilon}.
\end{aligned}$$

We then argue that the following two inequalities are true with high probability

$$\left|2n^{-1}\{(\boldsymbol{I} - \boldsymbol{P}_{X_S})\boldsymbol{X}_{S^*\backslash S}\boldsymbol{\beta}^*_{S^*\backslash S}\}^\top\boldsymbol{\epsilon}\right| < \frac{1}{4}\boldsymbol{\beta}^{*\top}_{S^*\backslash S}\hat{\boldsymbol{D}}(S)\boldsymbol{\beta}^*_{S^*\backslash S}, \tag{A.18}$$

$$n^{-1}\boldsymbol{\epsilon}^\top(\boldsymbol{P}_{X_S} - \boldsymbol{P}_{X_{S^*}})\boldsymbol{\epsilon} < \frac{1}{4}\boldsymbol{\beta}^{*\top}_{S^*\backslash S}\hat{\boldsymbol{D}}(S)\boldsymbol{\beta}^*_{S^*\backslash S}, \tag{A.19}$$

so that $n^{-1}(\mathcal{R}_{ols}(S, \mathcal{D}) - \mathcal{R}_{ols}(S^*, \mathcal{D})) > \frac{1}{2}\boldsymbol{\beta}^{*\top}_{S^*\backslash S}\hat{\boldsymbol{D}}(S)\boldsymbol{\beta}^*_{S^*\backslash S}$.

Defining $\boldsymbol{u}_S = n^{-1/2}(\boldsymbol{I} - \boldsymbol{P}_{X_S})\boldsymbol{X}_{S^*\backslash S}\boldsymbol{\beta}^*_{S^*\backslash S}$, we have that (A.18) is equivalent to

$$\frac{|\boldsymbol{u}_S^\top\boldsymbol{\epsilon}|}{\|\boldsymbol{u}_S\|_2} \leq \frac{n^{1/2}}{8}\|\boldsymbol{u}_S\|_2.$$

Note that since each of the entries of $\epsilon$ are i.i.d. zero-mean sub-Gaussian random variables with parameter $\sigma$, we have that $\frac{|\boldsymbol{u}_S^\top \epsilon|}{\|\boldsymbol{u}_S\|_2}$ is sub-Gaussian with parameter $\sigma$, so we can apply the Hoeffding bound (Proposition 2.5 in [34]) with $t = \sigma x$ to get that, for any $x > 0$,

$$\mathbb{P}\left(\frac{|\boldsymbol{u}_S^\top \epsilon|}{\|\boldsymbol{u}_S\|_2} > \sigma x\right) \le 2e^{-x^2/2}.$$

Now, applying union bound over all $S \in \mathcal{A}_t$, we have that for any $\xi > 0$,

$$\mathbb{P}\left(\exists S \in \mathcal{A}_t, \frac{|\boldsymbol{u}_S^\top \epsilon|}{\|\boldsymbol{u}_S\|_2} > \sigma\sqrt{\xi ts}\right) \le \binom{p-s}{t}\binom{s}{t} 2e^{-\xi ts/2} \le 2p^{2t}e^{-\xi ts/2}.$$

Then we have that, whenever

$$\frac{\inf_{S \in \mathcal{A}_t} \|\boldsymbol{u}_S\|_2}{t^{1/2}} \ge 8\sigma\left(\frac{\xi s}{n}\right)^{1/2},$$

we have that

$$\mathbb{P}\left(\exists S \in \mathcal{A}_t, \frac{|\boldsymbol{u}_S^\top \epsilon|}{\|\boldsymbol{u}_S\|_2} > \frac{n^{1/2}}{8}\|\boldsymbol{u}_S\|_2\right) \le 2p^{2t}e^{-\xi ts/2}.$$

Regarding (A.19), observe that, as shown in the proof of Theorem 2.1 of [13], we have that there exists a universal constant $c_1 > 0$ such that, for any $x > 0$,

$$\mathbb{P}\left(\frac{1}{n}|\epsilon^\top(\boldsymbol{P}_{X_S} - \boldsymbol{P}_{X_{S^*}})\epsilon| > \frac{2\sigma^2 x}{n}\right) \le 4e^{-c_1 \min\{x^2/t, x\}}.$$

Then, we have that for $c = \min\{c_1, \frac{1}{2}\}$,

$$\mathbb{P}\left(\frac{1}{n}|\epsilon^\top(\boldsymbol{P}_{X_S} - \boldsymbol{P}_{X_{S^*}})\epsilon| > \frac{2\sigma^2 x}{n}\right) \le 4e^{-c_1 \min\{x^2/t, x\}} \le 4e^{-c \min\{x^2/t, x\}}.$$

Noting that $s \ge 1$, we have that for any $\xi \ge 1$,

$$\mathbb{P}\left(\frac{1}{n}\epsilon^\top(\boldsymbol{P}_{X_S} - \boldsymbol{P}_{X_{S^*}})\epsilon > \frac{2\sigma^2 \xi ts}{n}\right) \le 4e^{-c\xi ts},$$

and applying union bound over $S \in \mathcal{A}_t$, we have

$$\mathbb{P}\left(\exists S \in \mathcal{A}_t, \frac{1}{n}\epsilon^\top(\boldsymbol{P}_{X_S} - \boldsymbol{P}_{X_{S^*}})\epsilon > \frac{2\sigma^2 \xi ts}{n}\right) \le \binom{p-s}{t}\binom{s}{t} 4e^{-c\xi ts} \le 4p^{2t}e^{-c\xi ts}.$$

Hence, whenever,

$$\frac{\inf_{S \in \mathcal{A}_t} \|\boldsymbol{u}_S\|_2}{t^{1/2}} \ge \left(\frac{8\xi\sigma^2 s}{n}\right)^{1/2},$$

we have that

$$\mathbb{P}\left(\exists S \in \mathcal{A}_t, \frac{1}{n}\epsilon^\top(\boldsymbol{P}_{X_S} - \boldsymbol{P}_{X_{S^*}})\epsilon > \frac{1}{4}\|\boldsymbol{u}_S\|_2^2\right) \le 4p^{2t}e^{-c\xi ts}.$$

Combining and taking union bound over all $t \in [s]$, we have that, for any $\xi > 1$, whenever

$$\tau \ge (8\sigma)^2 \frac{\xi s}{n},$$

we have that,

$$\mathbb{P}\left(\forall S \in \cup_{t=1}^s \mathcal{A}_t, \mathcal{R}_{ols}(S, \mathcal{D}) - \mathcal{R}_{ols}(S^*, \mathcal{D}) > \frac{1}{2}n\tau\right) \ge 1 - 4sp^{2s}(e^{-\xi s/2} + e^{-c\xi s}).$$

Now note that, since $c \le \frac{1}{2}$, we have that $1 - 4sp^{2s}(e^{-\xi s/2} + e^{-c\xi s}) \ge 1 - 8sp^{2s}e^{-c\xi s}$. Furthermore, choosing $\xi > \frac{2}{c}\log p$, we have that $p^{2s}e^{-c\xi s} = e^{2s\log p - c\xi s} \to 0$ as $p \to \infty$.

Combining the above, define $c_0 = \frac{4}{c} + b_y^2 + b_x^2 r^2$, where $r \geq (\frac{\kappa_+}{\kappa_-})M + 4\frac{\sigma b_x}{\kappa_-}$ as assumed, and let $C = \max\{8^2 c_0, \frac{8^2 c_0}{\sigma^2}\}$. Set $\xi = c_0 \log p$. Then, whenever

$$\tau \geq C\sigma^2 \frac{s \log p}{n},$$

we have that,

$$\mathbb{P}\left(\forall S \in \cup_{t=1}^s \mathcal{A}_t, \mathcal{R}_{ols}(S, \mathcal{D}) - \mathcal{R}_{ols}(S^*, \mathcal{D}) > \frac{1}{2}n\tau\right) \geq 1 - 8sp^{-\lambda s},$$

where $\lambda = cc_0 - 2 > 2$. Note that,

$$C\sigma^2 \frac{s \log p}{n} \geq C\frac{s}{n} \geq (8b_y^2 + 8b_x^2 r^2)\frac{s}{n} \geq 4\frac{\Delta}{n}$$

where the first inequality uses that $\max\{1, \frac{1}{\sigma^2}\}\sigma^2 \geq 1$, second inequality uses that $\log p > 1$ by assumption 4, and the last inequality follows from Lemma A.1. Hence, we have that

$$\mathbb{P}\left(\forall S \in \cup_{t=1}^s \mathcal{A}_t, \mathcal{R}_{ols}(S, \mathcal{D}) - \mathcal{R}_{ols}(S^*, \mathcal{D}) > 2\Delta\right) \geq 1 - 8sp^{-2}.$$

Taking union bound using Lemma A.7, we have that

$$\mathbb{P}\left(\forall S \in \cup_{t=1}^s \mathcal{A}_t, \mathcal{R}(S, \mathcal{D}) - \mathcal{R}(S^*, \mathcal{D}) > 2\Delta\right) \geq 1 - 2p^{-7} - 8sp^{-2} \geq 1 - 10sp^{-2}$$

and noting

$$\mathbb{P}(\mathcal{R}(\hat{S}_2(\mathcal{D}), \mathcal{D}) - \mathcal{R}(\hat{S}_1(\mathcal{D}), \mathcal{D}) > 2\Delta) \geq \mathbb{P}\left(\forall S \in \cup_{t=1}^s \mathcal{A}_t, \mathcal{R}(S, \mathcal{D}) - \mathcal{R}(S^*, \mathcal{D}) > 2\Delta\right)$$

concludes the proof. $\qquad\square$

### A.6 Proof of Theorem 3 and 4

Before proving the theorems, we will first setup some preliminaries.

Observe that the solution to the unconstrained least squares problem with support restricted to $S$ is given by

$$\boldsymbol{\beta}_{S,ols} = (\boldsymbol{X}_S^T \boldsymbol{X}_S)^{-1} \boldsymbol{X}_S^T \boldsymbol{y} = \underbrace{(\frac{\boldsymbol{X}_S^\top \boldsymbol{X}_S}{n})^{-1} \frac{\boldsymbol{X}_S^\top \boldsymbol{X}_{S^*} \boldsymbol{\beta}_{S^*}^*}{n}}_{:=\boldsymbol{u}_1} + \underbrace{(\frac{\boldsymbol{X}_S^\top \boldsymbol{X}_S}{n})^{-1} \frac{\boldsymbol{X}_S^\top \boldsymbol{\epsilon}}{n}}_{:=\boldsymbol{u}_2}$$

and the constrained estimator on the same support is given by

$$\boldsymbol{\beta}_{S,r} = \arg\min_{\boldsymbol{\beta}: \|\boldsymbol{\beta}\|_2 \leq r} \|\boldsymbol{y} - \boldsymbol{X}_S \boldsymbol{\beta}\|_2^2. \tag{A.20}$$

For each support $S$, we define the event $\mathcal{E}_{S,r} := \{\boldsymbol{\beta}_{S,r} = \boldsymbol{\beta}_{S,ols}\}$ and the intersection of events across all supports as $\mathcal{E}_r = \cap_{S:|S|=s}\mathcal{E}_{S,r}$.

The lemma below shows that if a sufficiently high bound on the $\ell_2$ norm of $\boldsymbol{\beta}$ in the constrained optimization in A.20 is chosen, the solution to the unconstrained OLS problem is the same as the solution to A.20 for *all* supports $S$ with high probability.

**Lemma A.7.** *Suppose assumptions 1-4 hold. If $r \geq (\frac{\kappa_+}{\kappa_-})M + 4\frac{\sigma b_x}{\kappa_-}$ then $\mathbb{P}[\mathcal{E}_r] = \mathbb{P}[\cap_{S:|S|=s}\mathcal{E}_{S,r}] \geq 1 - 2p^{-7}$.*

*Proof.* At a high level, we are seeking to bound $\|\boldsymbol{\beta}_{S,ols}\|_2$ with high probability. First off, by assumption 2 and 3, we have that $\|\boldsymbol{u}_1\|_2 \leq \|(\boldsymbol{X}_S^\top \boldsymbol{X}_S/n)^{-1}\|_2 \|\boldsymbol{X}_S^\top \boldsymbol{X}_{S^*}/n\|_2 \|\boldsymbol{\beta}_{S^*}^*\|_2 \leq (\frac{\kappa_+}{\kappa_-})M$. Next, note that

$$\|\boldsymbol{u}_2\|_2 \leq \left\|(\frac{\boldsymbol{X}_S^\top \boldsymbol{X}_S}{n})^{-1}\right\|_2 \left\|\frac{\boldsymbol{X}_S^\top \boldsymbol{\epsilon}}{n}\right\|_2 \leq \sqrt{s}\left\|(\frac{\boldsymbol{X}_S^\top \boldsymbol{X}_S}{n})^{-1}\right\|_2 \left\|\frac{\boldsymbol{X}_S^\top \boldsymbol{\epsilon}}{n}\right\|_\infty$$

$$\leq \sqrt{s}\left\|(\frac{\boldsymbol{X}_S^\top \boldsymbol{X}_S}{n})^{-1}\right\|_2 \left\|\frac{\boldsymbol{X}^\top \boldsymbol{\epsilon}}{n}\right\|_\infty.$$

Hence, we have $\|\boldsymbol{u}_2\|_2 \leq \frac{\sqrt{s}}{\kappa_-} \left\| \frac{\boldsymbol{X}^\top \boldsymbol{\epsilon}}{n} \right\|_\infty$. Now, define $D_{i,j} = X_{i,j}\epsilon_j$ for all $(i,j) \in [p] \times [n]$. Since $\epsilon_j$ is sub-Gaussian with parameter $\sigma$, using assumption 1 we have that $D_{i,j}$ is sub-Gaussian with parameter $\sigma b_x$. Applying the Hoeffding bound (Proposition 2.5 in [34]) with $t = 4\sigma b_x\sqrt{n \log p}$, we have that, for all $i \in [p]$,

$$\mathbb{P}\left[\frac{1}{n}|D_{i,j}| \geq 4\sigma b_x \sqrt{\frac{\log p}{n}}\right] \leq 2p^{-8}.$$

Observe that $\left\| \frac{\boldsymbol{X}^\top \boldsymbol{\epsilon}}{n} \right\|_\infty = \max_{i \in [p]} \frac{1}{n}|D_{i,j}|$. Hence, by union bound, we have that $\mathbb{P}\left[\left\| \frac{\boldsymbol{X}^\top \boldsymbol{\epsilon}}{n} \right\|_\infty \geq 4\sigma b_x \sqrt{\frac{\log p}{n}}\right] \leq 2p^{-7}$. By assumption 4 we have that

$$\mathbb{P}\left[\left\| \frac{\boldsymbol{X}^\top \boldsymbol{\epsilon}}{n} \right\|_\infty \geq 4\frac{\sigma b_x}{\sqrt{s}}\right] \leq 2p^{-7}$$

This yields that $\|\beta_{S,ols}\|_2 \leq \|\boldsymbol{u}_1\|_2 + \|\boldsymbol{u}_2\|_2 \leq (\frac{\kappa_+}{\kappa_-})M + 4\frac{\sigma b_x}{\kappa_-}$ with probability at least $1 - 2p^{-7}$. Hence, we have that if $r \geq (\frac{\kappa_+}{\kappa_-})M + 4\frac{\sigma b_x}{\kappa_-}$ then $\mathbb{P}[\mathcal{E}_r] \geq 1 - 2p^{-7}$ as desired. $\qquad \square$

For the remainder of the paper, define $\mathcal{R}_{ols}(S, \mathcal{D}) = \min_{\boldsymbol{\beta} \in \mathbb{R}^s} \|\boldsymbol{y} - \boldsymbol{X}_S\boldsymbol{\beta}\|_2^2$ for all $S \subset [p]$ such that $|S| = s$. Define the event $\mathcal{E}_{gap} := \bigcap_{t=1}^s \{\forall S \in \mathcal{A}_t, \ \frac{1}{n}(\mathcal{R}_{ols}(S, \mathcal{D}) - \mathcal{R}_{ols}(S^*, \mathcal{D})) \geq \frac{1}{2}t\tau\}$.

**Lemma A.8.** *Suppose $p \geq 3$. There exists a universal constant $C > 0$ such that, whenever*

$$\tau \geq C\sigma^2 \frac{\log p}{n}$$

*we have that*

$$\mathbb{P}[\mathcal{E}_{gap}] = \mathbb{P}\left[\bigcap_{t=1}^s \{\forall S \in \mathcal{A}_t, \ \frac{1}{n}(\mathcal{R}_{ols}(S, \mathcal{D}) - \mathcal{R}_{ols}(S^*, \mathcal{D})) \geq \frac{1}{2}t\tau\}\right] \geq 1 - 8sp^{-2}.$$

*Proof.* By Theorem 2.1 of [13], we have that there exists constant $c > 0$ such that for any $\xi > 1$, whenever

$$\frac{\min_{S \in \mathcal{A}_t} \boldsymbol{\beta}_{S^*\setminus S}^{*\top} \hat{\boldsymbol{D}}(S)\boldsymbol{\beta}_{S^*\setminus S}^*}{t} \geq \left(\frac{4\xi}{1-\eta}\right)^2 \frac{\sigma^2 \log p}{n},$$

we have

$$\mathbb{P}\left[\forall S \in \mathcal{A}_t, \frac{1}{n}(\mathcal{R}_{ols}(S, \mathcal{D}) - \mathcal{R}_{ols}(S^*, \mathcal{D})) > \eta\boldsymbol{\beta}_{S^*\setminus S}^{*\top} \hat{\boldsymbol{D}}(S)\boldsymbol{\beta}_{S^*\setminus S}^*\right] \geq 1 - 4p^{-(c\xi-2)t} - 2p^{-(\xi^2-2)t}.$$

Let $\eta = \frac{1}{2}, \xi = \max\{2, \frac{4}{c}\}$, and $C = \left(\frac{4\xi}{1-\eta}\right)^2$. Then, we have that $-(c\xi - 2)t \leq -2t$ and $-(\xi^2 - 2)t \leq -2t$, so $1 - 4p^{-(c\xi-2)t} - 2p^{-(\xi^2-2)t} \geq 1 - 4p^{-2t} - 2p^{-2t} \geq 1 - 8p^{-2t}$. Observe also that

$$\frac{\min_{S \in \mathcal{A}_t} \boldsymbol{\beta}_{S^*\setminus S}^{*\top} \hat{\boldsymbol{D}}(S)\boldsymbol{\beta}_{S^*\setminus S}^*}{t} \geq \min_{t \in [s]} \frac{\min_{S \in \mathcal{A}_t} \boldsymbol{\beta}_{S^*\setminus S}^{*\top} \hat{\boldsymbol{D}}(S)\boldsymbol{\beta}_{S^*\setminus S}^*}{t} = \tau.$$

Hence, we have that whenever

$$\tau \geq C\sigma^2 \frac{\log p}{n},$$

we have

$$\mathbb{P}\left[\forall S \in \mathcal{A}_t, \frac{1}{n}(\mathcal{R}_{ols}(S, \mathcal{D}) - \mathcal{R}_{ols}(S^*, \mathcal{D})) > \frac{1}{2}\boldsymbol{\beta}_{S^*\setminus S}^{*\top} \hat{\boldsymbol{D}}(S)\boldsymbol{\beta}_{S^*\setminus S}^*\right] \geq 1 - 8p^{-2t}.$$

Observe now that $\boldsymbol{\beta}_{S^*\setminus S}^{*\top} \hat{\boldsymbol{D}}(S)\boldsymbol{\beta}_{S^*\setminus S}^* \geq t\tau$ hence we have

$$\mathbb{P}\left[\forall S \in \mathcal{A}_t, \frac{1}{n}(\mathcal{R}_{ols}(S, \mathcal{D}) - \mathcal{R}_{ols}(S^*, \mathcal{D})) \geq \frac{1}{2}t\tau\right] \geq 1 - 8p^{-2t}.$$

Applying the union bound and the fact that $t \geq 1$, we get $\mathbb{P}[\mathcal{E}_{gap}] \geq 1 - 8sp^{-2}$, which concludes the proof.

$\qquad \square$

We now proceed with the proof of the theorems.

***Proof of Theorem 3.*** We will now show that the exponential mechanism with scoring function $\hat{\mathcal{R}}$ and $R = 2$, denoted as $\hat{\mathcal{A}}_{E_2}$, recovers the true support with high probability.

We first define the event

$$\mathcal{E} := \bigcap_{t=1}^{s} \{\forall S \in \mathcal{A}_t, \mathcal{R}(S, \mathcal{D}) - \mathcal{R}(S^*, \mathcal{D}) \geq \frac{1}{2} nt\tau\}.$$

Observe that $\mathcal{E}_r \cap \mathcal{E}_{\text{gap}} \subset \mathcal{E}$. By Lemmas A.7 and A.8, if we apply union bound, we have that $\mathbb{P}[\mathcal{E}] \geq \mathbb{P}[\mathcal{E}_r \cap \mathcal{E}_{\text{gap}}] \geq 1 - 8sp^{-2} - 2p^{-7} \geq 1 - 10sp^{-2}$. Furthermore, if we condition on $\mathcal{E}$, we have that $\mathcal{R}(\hat{S}_1(\mathcal{D}), \mathcal{D}) = \mathcal{R}(S^*, \mathcal{D})$.

Then, note that

$$\mathbb{P}(\hat{\mathcal{A}}_{E_2}(\mathcal{D}) = S^* | \mathcal{E}) = \frac{1}{1 + (\binom{p}{s} - 1) \exp(-\frac{\epsilon}{2\Delta}(\mathcal{R}(\hat{S}_2(\mathcal{D}), \mathcal{D}) - \mathcal{R}(\hat{S}_1(\mathcal{D}), \mathcal{D})))}).$$

Then, we have that, if we assume

$$\tau \geq \max\{C\sigma^2, \frac{8\Delta}{\epsilon}s\} \frac{\log p}{n},$$

we have the following:

$$(\binom{p}{s} - 1) \exp(-\frac{\epsilon}{2\Delta} \underbrace{(\mathcal{R}(\hat{S}_2(\mathcal{D}), \mathcal{D}) - \mathcal{R}(\hat{S}_1(\mathcal{D}), \mathcal{D}))}_{:=G}) \leq p^s \exp(-\frac{\epsilon G}{2\Delta})$$

$$\leq p^s \exp(-\frac{\epsilon n\tau}{4\Delta})$$

$$\leq p^s \exp(-2s \log p)$$

$$= p^{-s}$$

where the second inequality uses the fact that $t \geq 1$ for all $S \in \cup_{t=1}^{s} \mathcal{A}_t$. Thus, we have

$$\mathbb{P}(\hat{\mathcal{A}}_{E_2}(\mathcal{D}) = S^* | \mathcal{E}) \geq \frac{1}{1 + p^{-s}}.$$

Now consider any exponential mechanism with scoring function $\hat{\mathcal{R}}$ and $R > 2$, denoted $\hat{\mathcal{A}}_E(\mathcal{D})$. By the same argument used in Lemma A.5, we have that

$$\mathbb{P}(\hat{\mathcal{A}}_{E_2}(\mathcal{D}) = S^* | \mathcal{E}) \leq \mathbb{P}(\hat{\mathcal{A}}_E(\mathcal{D}) = S^* | \mathcal{E}).$$

Applying the law of total probability, we have that

$$\mathbb{P}(\hat{\mathcal{A}}_E(\mathcal{D}) = S^*) = \mathbb{P}(\mathcal{E})\mathbb{P}(\hat{\mathcal{A}}_E(\mathcal{D}) = S^* | \mathcal{E}) + \mathbb{P}(\mathcal{E}^c)\mathbb{P}(\hat{\mathcal{A}}_E(\mathcal{D}) = S^* | \mathcal{E}^c)$$

$$\geq \mathbb{P}(\mathcal{E})\mathbb{P}(\hat{\mathcal{A}}_E(\mathcal{D}) = S^* | \mathcal{E})$$

$$\geq \frac{1 - 10sp^{-2}}{1 + p^{-s}}$$

Now we apply Lemma A.4. Let $q = \frac{R}{\binom{p}{s}}$. Then, we have that $(1 - q^T)\mathbb{P}(\hat{\mathcal{A}}_E(\mathcal{D}) = S^*) \leq \mathbb{P}(\hat{\mathcal{M}}(\mathcal{D}) = S^*)$, and as $T \to \infty$, we have that $\mathbb{P}(\hat{\mathcal{A}}_E(\mathcal{D}) = S^*) \leq \mathbb{P}(\hat{\mathcal{M}}(\mathcal{D}) = S^*)$, and we conclude that

$$\mathbb{P}(\hat{\mathcal{M}}(\mathcal{D}) = S^*) \geq \frac{1 - 10sp^{-2}}{1 + p^{-s}}.$$

$\square$

***Proof of Theorem 4.*** Let $\Delta$ be the bounded global sensitivity of $\mathcal{R}$ as in Lemma A.1. First, define the event $\mathcal{E}_{2\Delta} = \{\forall S \in \cup_{t=1}^s \mathcal{A}_t, \mathcal{R}_{ols}(S, \mathcal{D}) - \mathcal{R}_{ols}(S^*, \mathcal{D}) > 2\Delta\}$. In the proof of Lemma A.6, we show that, given assumptions 1-4, whenever

$$\tau \geq C\sigma^2 \frac{s \log p}{n},$$

we have that

$$\mathbb{P}\left(\forall S \in \cup_{t=1}^s \mathcal{A}_t, \mathcal{R}_{ols}(S, \mathcal{D}) - \mathcal{R}_{ols}(S^*, \mathcal{D}) > 2\Delta\right) \geq 1 - 8sp^{-2}.$$

Now consider the event

$$\mathcal{E} := \left(\bigcap_{t=1}^s \left\{\forall S \in \mathcal{A}_t, \ \mathcal{R}(S, \mathcal{D}) - \mathcal{R}(S^*, \mathcal{D}) \geq \frac{1}{2} n t \tau\right\}\right)$$
$$\cap \left\{\max_{\substack{S \subseteq [p] \\ |S|=s}} \max_{\substack{\mathcal{D}, \mathcal{D}' \in \mathcal{Z}^n \\ \mathcal{D}, \mathcal{D}' \text{ neighbors}}} \tilde{\mathcal{R}}(S, \mathcal{D}) - \tilde{\mathcal{R}}(S, \mathcal{D}') \leq \Delta\right\}.$$

By Theorem 2, we have that if $\mathcal{R}(S, \mathcal{D}) - \mathcal{R}(S^*, \mathcal{D}) > 2\Delta$ for all $S \in \cup_{t=1}^s \mathcal{A}_t$, which implies that $\mathcal{R}(\hat{S}_2(\mathcal{D}), \mathcal{D}) - \mathcal{R}(\hat{S}_1(\mathcal{D}), \mathcal{D}) > 2\Delta$, then $\tilde{\mathcal{R}}$ has the same bound on global sensitivity as $\mathcal{R}$ in Lemma A.1. Hence, we have that $\mathcal{E}_r \cap \mathcal{E}_{\text{gap}} \cap \mathcal{E}_{2\Delta} \subset \mathcal{E}$. Then, by Lemmas A.7 and A.8, if we apply union bound, we have that $\mathbb{P}[\mathcal{E}] \geq \mathbb{P}[\mathcal{E}_r \cap \mathcal{E}_{\text{gap}} \cap \mathcal{E}_{2\Delta}] \geq 1 - 8sp^{-2} - 2p^{-7} - 8sp^{-2} \geq 1 - 18sp^{-2}$. Furthermore, if we condition on $\mathcal{E}$, we have that $\mathcal{R}(\tilde{S}_0(\mathcal{D}), \mathcal{D}) = \mathcal{R}(S^*, \mathcal{D})$.

Then, note that

$$\mathbb{P}[\tilde{\mathcal{M}}(\mathcal{D}) = S^*|\mathcal{E}] = \frac{1}{1 + \sum_{t=1}^s \binom{p-s}{t}\binom{s}{t} \exp(\frac{-\epsilon(\mathcal{R}(\tilde{S}_t(\mathcal{D}),\mathcal{D})-\mathcal{R}(\tilde{S}_0(\mathcal{D}),\mathcal{D}))}{2\Delta})}.$$

Then, we have that, if we assume

$$\tau \geq \max\{C\sigma^2 s, \frac{16\Delta}{\epsilon}\}\frac{\log p}{n},$$

we have the following:

$$\sum_{t=1}^s \binom{p-s}{t}\binom{s}{t} \exp(\frac{-\epsilon(\mathcal{R}(\tilde{S}_t(\mathcal{D}),\mathcal{D})-\mathcal{R}(\tilde{S}_0(\mathcal{D}),\mathcal{D}))}{2\Delta}) \leq \sum_{t=1}^s p^{2t} \exp(\frac{-\epsilon nt\tau}{4\Delta})$$
$$\leq \sum_{t=1}^s p^{2t} p^{-4t}$$
$$\leq \sum_{t=1}^s p^{-2t} \leq 2p^{-2}$$

where we use assumption 4 in the last inequality. Thus, we have that

$$\mathbb{P}[\tilde{\mathcal{M}}(\mathcal{D}) = S^*|\mathcal{E}] \geq \frac{1}{1 + 2p^{-2}}.$$

Thus, we conclude that

$$\mathbb{P}(\tilde{\mathcal{M}}(\mathcal{D}) = S^*) = \mathbb{P}(\mathcal{E})\mathbb{P}(\tilde{\mathcal{M}}(\mathcal{D}) = S^*|\mathcal{E}) + \mathbb{P}(\mathcal{E}^c)\mathbb{P}(\tilde{\mathcal{M}}(\mathcal{D}) = S^*|\mathcal{E}^c)$$
$$\geq \mathbb{P}(\mathcal{E})\mathbb{P}(\tilde{\mathcal{M}}(\mathcal{D}) = S^*|\mathcal{E})$$
$$\geq \frac{1 - 18sp^{-2}}{1 + 2p^{-2}}.$$

$\square$

### A.7 Proof of Proposition 1

*Proof.* Take $k \geq 1$. Let $S$ and $(z, \beta, \theta)$ be feasible for Problems 5 and 9 equivalently. Observe that, since $z \in \{0,1\}^p$ and $\sum_{i=1}^p z_i = s$, we have $|\{i : z_i \neq 0\}| = s$. Furthermore, the constraint that $\sum_{i \in \hat{S}_j(\mathcal{D})} z_i \leq s - \frac{1}{2} < s \; \forall j \in [k-1]$ implies that $\{i : z_i \neq 0\} \neq \hat{S}_i(\mathcal{D}) \; \forall i \in [k-1]$. Finally, observe that, from the constraint $\beta_i^2 \leq \theta_i z_i \; \forall i \in [p]$ it follows that $\beta$ can be nonzero only on indices $i$ for which $z_i = 1$, and combined with the constraint $\sum_{i=1}^p \theta_i \leq r^2$, we have that $\|\beta\|_2^2 \leq r^2$. We then have that Problem 9 solves the problem of minimizing $\|y - X\beta\|_2^2$ among all $\beta$ such that $\|\beta\|_2^2 \leq r^2$, and such that $\text{supp}(\beta) \subset \{i : z_i \neq 0\}$. This then becomes analogous to restricting $X$ to the columns indexed by $\{i : z_i \neq 0\}$ and we have that the optimization formulations in Problems 5 and 9 are exactly equivalent, with $S = \{i : z_i \neq 0\}$. $\qquad \square$

## B    BSS algorithmic details

Experimentally, we make a number of adjustments to Algorithm 2 to facilitate a faster process of obtaining $\hat{S}_k(\mathcal{D})$ for $k \in [R]$. We find that, very commonly in simulated experiments, the enumerated supports $\hat{S}_2(\mathcal{D}), ..., \hat{S}_{1+(p-s)s}(\mathcal{D})$ are the $(p-s)s$ supports that make 1 mistake from $\hat{S}_1(\mathcal{D})$, and that the largest gaps in objective value across two consecutive enumerated supports often occur when the consecutive supports belong to different elements of the partition $P_0(\mathcal{D}), P_1(\mathcal{D}), P_2(\mathcal{D}), ..., P_s(\mathcal{D})$. Thus, in order to make the best choice of $R$ to explore the objective landscape better but without significantly increasing computational costs, we pursued the following strategy:

1. We use Algorithm 2 to obtain $\hat{S}_1(\mathcal{D}) = \tilde{S}_0(\mathcal{D})$.

2. We solve $c(z_k) = \min_{\|\beta\|_2^2 \leq r^2} \frac{1}{2n} \|y - X\beta\|_2^2 + \frac{\lambda}{2n} \sum_{i=1}^p \frac{\beta_i^2}{(z_k)_i}$ for all $(p-s)s$ binary vectors $z_k$ corresponding to the supports in $P_1(\mathcal{D})$.

3. We use Algorithm 2 to obtain the optimal support that makes at least 2 mistakes from $\tilde{S}_0(\mathcal{D})$, with corresponding binary vector $\tilde{z}_2$.

4. We check that $\max_{k \in [(p-s)s]} c(z_k) \leq c(\tilde{z}_2)$.

5. We run Algorithm 1 with (a) the optimal support, (b) all the supports with 1 mistake, and (c) the optimal with 2 mistakes, i.e. $R = 2 + (p-s)s$.

Furthermore, to reduce the number of iterations of outer approximation needed in step 3 above, we added additional cuts corresponding to some of the 1-mistake vectors $z_k$ prior to the start of the while loop in Algorithm 2. We selected cuts by first sorting the values of $X^\top y$ in absolute value, then taking the top $m\%$ of the features from this sorting, and using those entries to make two swaps to the binary vector corresponding to $\hat{S}_1(\mathcal{D})$, thus generating 1-mistake vectors that are then used for cuts.

### B.1    Solving $c(\hat{z})$ using PGD

At each iteration of outer approximation in Algorithm 2, we use projected gradient descent (PGD) to solve

$$c(\hat{z}) = \min_{\|\beta\|_2^2 \leq r^2} \underbrace{\frac{1}{2n} \|y - X\beta\|_2^2 + \frac{\lambda}{2n} \sum_{i=1}^p \frac{\beta_i^2}{\hat{z}_i}}_{g(\beta)}.$$

We have that

$$\nabla g(\beta) = \frac{1}{n} X^T (X\beta - y) + \frac{\lambda}{n} \frac{\beta}{\hat{z}} = \frac{1}{n} (X^\top (X\beta - y) + \lambda \text{Diag}(\frac{1}{\hat{z}_1}, ..., \frac{1}{\hat{z}_p})\beta)$$

where $\frac{v}{u}$ for vectors $v, u \in \mathbb{R}^p$ denotes element-wise division. Then note that

$$||\frac{1}{n}(X^\top(X\beta - y) + \lambda\text{Diag}(\frac{1}{\hat{z}_1}, ..., \frac{1}{\hat{z}_p})\beta) - \frac{1}{n}(X^\top(X\beta' - y) + \lambda\text{Diag}(\frac{1}{\hat{z}_1}, ..., \frac{1}{\hat{z}_p})\beta')||_2$$

$$= ||\frac{1}{n}(X^\top X + \lambda\text{Diag}(\frac{1}{\hat{z}_1}, ..., \frac{1}{\hat{z}_p}))(\beta - \beta')||_2$$

$$\leq \frac{1}{n}\lambda_{\max}(X^\top X + \lambda\text{Diag}(\frac{1}{\hat{z}_1}, ..., \frac{1}{\hat{z}_p}))||\beta - \beta'||_2$$

Then setting $L = \frac{1}{n}\lambda_{\max}(X^\top X + \lambda\text{Diag}(\frac{1}{\hat{z}_1}, ..., \frac{1}{\hat{z}_p}))$, the PGD update is then

$$\beta_{t+1} = \frac{r(\beta_t - \frac{1}{L}\nabla g(\beta_t))}{\max\{r, ||\beta_t - \frac{1}{L}\nabla g(\beta_t)||_2\}}.$$

### B.2 Heuristic to kickstart outer approximation

In order to provide a good initialization for $z_0$ in Algorithm 2, we use the heuristic taken from Algorithm 1 in [5]. Specifically, we consider the problem

$$\min_{\beta \in \mathbb{R}^p : ||\beta||_0 \leq s} \underbrace{\frac{1}{2n}(||y - X\beta|| + \lambda||\beta||_2^2)}_{h(\beta)}.$$

Note that $\nabla h(\beta) = \frac{1}{n}(X^\top(X\beta - y) + \lambda\beta)$ and that

$$||\frac{1}{n}(X^\top(X\beta - y) + \lambda\beta) - \frac{1}{n}(X^\top(X\beta' - y) + \lambda\beta')||_2 = ||\frac{1}{n}(X^\top X + \lambda I)(\beta - \beta')||_2$$

$$\leq \frac{1}{n}\lambda_{\max}(X^\top X + \lambda I)||\beta - \beta'||_2$$

Then, letting $L = \frac{1}{n}\lambda_{\max}(X^\top X + \lambda I)$, we run the following heuristic:

1. Initialize $\beta_1 \in \mathbb{R}^p$ such that $||\beta_1||_0 \leq s$.
2. For $t \geq 1$:
    (a) Sort the entries of $\beta_t - \frac{1}{L}\nabla h(\beta_t)$ in order of decreasing absolute value, let $I$ denote the index set of the $s$ largest entries.
    (b) Set $(\beta_{t+1})_i = (\beta_t - \frac{1}{L}\nabla h(\beta_t))_i$ if $i \in I$, and $(\beta_{t+1})_i = 0$ otherwise.

## C  Modifications pertaining to hinge loss

Similarly to BSS, we now consider the following sparse classification problem:

$$\min_{\beta \in \mathbb{R}^p} \frac{1}{n}\sum_{i=1}^n \max\{0, 1 - y_i(x_i^T\beta)\} \text{ s.t. } ||\beta||_0 \leq s, ||\beta||_2 \leq r \tag{C.1}$$

Our objective function then becomes

$$\mathcal{R}(S, \mathcal{D}) = \min_{\beta \in \mathbb{R}^{|S|}} \frac{1}{n}\sum_{i=1}^n \max\{0, 1 - y_i((x_i)_S^T\beta)\} \text{ s.t. } ||\beta||_2 \leq r$$

where $\mathcal{D} = (X, y)$ as before, and $(x_i)_S \in \mathbb{R}^s$ is the $i$-th row of $X$ with columns indexed by $S$.

We first present a result analogous to Lemma A.1 in order to bound the global sensitivity for the case of hinge loss.

**Lemma C.1.** *Suppose that* $|X_{i,j}| \leq b_x$ *for* $i \in [n], j \in [p]$. *Then,*

$$\Delta \leq \frac{1}{n}(1 + rb_x\sqrt{s})$$

*Proof.* Suppose $\mathcal{D}, \mathcal{D}'$ are two neighboring datasets. Fix a support $S \in \mathcal{O}$ and suppose

$$\hat{\boldsymbol{\beta}} \in \arg\min_{\|\boldsymbol{\beta}\|_2 \leq r} \frac{1}{n} \sum_{i=1}^{n} \max\{0, 1 - y_i'((\boldsymbol{x}_i')_S^T \boldsymbol{\beta})\}.$$

Then note that

$$\mathcal{R}(S, \mathcal{D}) - \mathcal{R}(S, \mathcal{D}') \leq \frac{1}{n} (\sum_{i=1}^{n} \max\{0, 1 - y_i((\boldsymbol{x}_i)_S^T \hat{\boldsymbol{\beta}})\} - \sum_{i=1}^{n} \max\{0, 1 - y_i'((\boldsymbol{x}_i')_S^T \hat{\boldsymbol{\beta}})\}).$$

Let us assume without loss of generality that $\mathcal{D}, \mathcal{D}'$ differ in the $n$-th observations. Then we have that

$$\frac{1}{n} (\sum_{i=1}^{n} \max\{0, 1 - y_i((\boldsymbol{x}_i)_S^T \hat{\boldsymbol{\beta}})\} - \sum_{i=1}^{n} \max\{0, 1 - y_i'((\boldsymbol{x}_i')_S^T \hat{\boldsymbol{\beta}})\})$$

$$= \frac{1}{n} (\max\{0, 1 - y_n((\boldsymbol{x}_n)_S^T \hat{\boldsymbol{\beta}})\} - \max\{0, 1 - y_n'((\boldsymbol{x}_n')_S^T \hat{\boldsymbol{\beta}})\})$$

$$\leq \frac{1}{n} \max\{0, 1 - y_n((\boldsymbol{x}_n)_S^T \hat{\boldsymbol{\beta}})\}$$

$$\leq \frac{1}{n} (1 + r b_x \sqrt{s})$$

where the last inequality uses the fact that $|S| = s$ and the Cauchy-Schwarz inequality, since
$-y_n((\boldsymbol{x}_n)_S^T \hat{\boldsymbol{\beta}}) \leq |y_n((\boldsymbol{x}_n)_S^T \hat{\boldsymbol{\beta}})| = |(\boldsymbol{x}_n)_S^T \hat{\boldsymbol{\beta}}| \leq \|(\boldsymbol{x}_n)_S\|_2 \|\hat{\boldsymbol{\beta}}\|_2$. □

## C.1 Optimization formulation

As in the BSS case, we consider the penalized form of Problem C.1 in order to obtain $\hat{S}_k(\mathcal{D})$ for $k \in [R]$. We define

$$c(\boldsymbol{z}) = \min_{\|\boldsymbol{\beta}\|_2^2 \leq r^2} \frac{1}{n} \sum_{i=1}^{n} \max\{0, 1 - y_i(\boldsymbol{x}_i^T \boldsymbol{\beta})\} + \frac{\lambda}{n} \sum_{i=1}^{p} \frac{\beta_i^2}{z_i}$$

and we seek to solve

$$\min_{\boldsymbol{z}} \quad c(\boldsymbol{z})$$

$$\text{subject to } \boldsymbol{z} \in \{0, 1\}^p, \ \sum_{i=1}^{p} z_i = s,$$

$$\sum_{i \in \hat{S}_j(\mathcal{D})} z_i \leq s - \frac{1}{2} \ \forall j \in [k-1].$$

Given $\boldsymbol{z} \in \{0, 1\}^p$, define $\hat{\boldsymbol{z}} \in (0, 1]^p$ as in Section 4. Let

$$\hat{\boldsymbol{\beta}} \in \arg\min_{\|\boldsymbol{\beta}\|_2^2 \leq r^2} \frac{1}{n} \sum_{i=1}^{n} \max\{0, 1 - y_i(\boldsymbol{x}_i^T \boldsymbol{\beta})\} + \frac{\lambda}{n} \sum_{i=1}^{p} \frac{\beta_i^2}{\hat{z}_i}.$$

We then have

$$(\nabla c(\hat{\boldsymbol{z}}))_i = -\frac{\lambda}{n} \frac{(\hat{\beta}_i)^2}{\hat{z}_i^2}$$

and we run Algorithm 2 with these modifications.

## C.2 Solving $c(\hat{\boldsymbol{z}})$ using Projected Subgradient method

At each iteration of outer approximation, we solve

$$\min_{\|\boldsymbol{\beta}\|_2^2 \leq r^2} \frac{1}{n} \sum_{i=1}^{n} \max\{0, 1 - y_i(\boldsymbol{x}_i^T \boldsymbol{\beta})\} + \frac{\lambda}{n} \sum_{i=1}^{p} \frac{\beta_i^2}{\hat{z}_i}$$

using a projected subgradient method. Using the subgradient

$$g(\boldsymbol{\beta}) = \frac{1}{n}\left(\sum_{\substack{i=1 \\ y_i\boldsymbol{\beta}^T\boldsymbol{x}_i<1}} -y_i\boldsymbol{x}_i + 2\lambda\text{Diag}(\frac{1}{\hat{z}_1},...,\frac{1}{\hat{z}_p})\boldsymbol{\beta}\right),$$

we run the update

$$\boldsymbol{\beta}_{t+1} = \frac{r(\boldsymbol{\beta}_t - \eta_t g(\boldsymbol{\beta}_t))}{\max\{r, ||\boldsymbol{\beta}_t - \eta_t g(\boldsymbol{\beta}_t)||_2\}}$$

where $\eta_t = \frac{1}{\sqrt{t}}$.

# D   Additional experimental results

## D.1   BSS

### D.1.1   Prediction accuracy and utility loss

In an effort to compare prediction accuracy across methods, we performed a 70/30 random train/test split and implemented Algorithm 2 in [23] on the training data, using half of the privacy budget (i.e., $\epsilon/2$) for variable selection with the top-$R$, mistakes, Samp-Agg, or MCMC methods, and the remaining half for model optimization via objective perturbation (Algorithm 1 in [23]) to obtain the regression coefficients under the privacy budget ($\beta_{\text{priv}}$). We ran experiments with $p = 100$, $s = 5$, $\epsilon = 2$, SNR $= 5$, and $\rho = 0.1$, and conducted 10 independent trials for each value of $n$. For each trial, we drew 100 samples from the distribution corresponding to each algorithm. For MCMC, we used 100 independent Markov chains per trial. After running objective perturbation over the selected supports, we obtained 1000 distinct coefficient vectors for each method and each $n$, and computed the average MSE on the test data. The choices of $\lambda = 120$ and MCMC iterations $= 1000$ were made to keep the runtimes comparable, as in our support recovery results. We summarize the results below, showing that our top-$R$ and mistakes methods outperform the competitor algorithms in prediction accuracy for sufficiently large $n$.

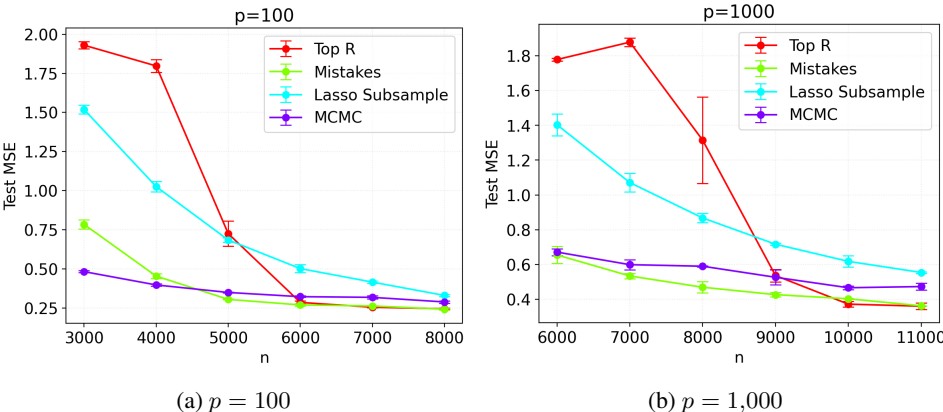

(a) $p = 100$                (b) $p = 1,000$

Figure D.1: Numerical experiments for $p = 100$ and $p = 1,000$, with $s = 5$, SNR=5, $\rho = 0.1$, and $\epsilon = 2$. The penalty parameter $\lambda$ in Algorithm 2 was set to 120 and 250 for figures D.1a and D.1b, respectively. On the $x$-axis, we vary the value of $n$ and plot the average test MSE across 10 independent trials. Error bars denote the mean standard error.

Furthermore, we ran experiments to evaluate the utility loss, defined as the gap between the objective at $\beta_{\text{priv}}$ and the objective at $\hat{\beta}$ (the optimal BSS solution), using the same parameters as in the experiments above. As with the prediction accuracy results, our top-$R$ and mistakes methods outperform the competitor algorithms in terms of utility loss when $n$ is large enough.

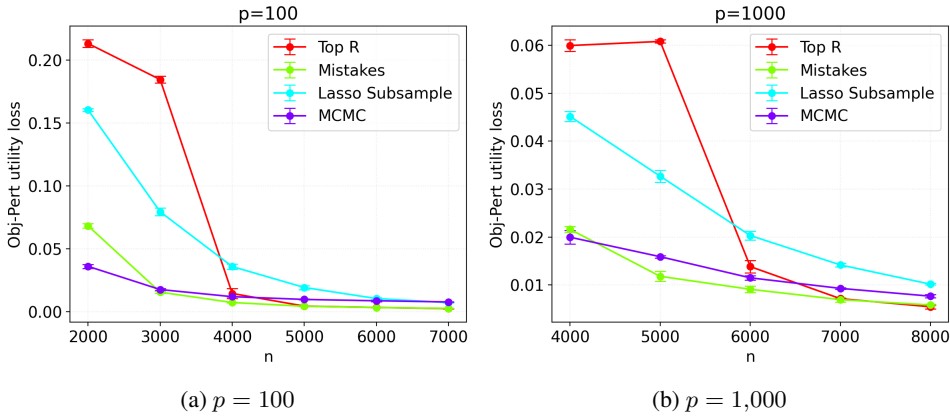

(a) $p = 100$          (b) $p = 1{,}000$

Figure D.2: Numerical experiments for $p = 100$ and $p = 1{,}000$, with $s = 5$, SNR=5, $\rho = 0.1$, and $\epsilon = 2$. The penalty parameter $\lambda$ in Algorithm 2 was set to 120 and 250 for figures D.2a and D.2b, respectively. On the $x$-axis, we vary the value of $n$ and plot the average utility loss across 10 independent trials. Error bars denote the mean standard error.

### D.1.2 Ablation studies

In this section, we present several ablation studies in order to show the effect of changing $R$, $\lambda$, and $(b_x, b_y)$ on the fraction of correctly recovered supports and on the F1 score. For the following results, we ran 10 independent trials and drew 100 samples from the distribution corresponding to each of our algorithms.

**Ablation study of $R$.** As noted in Appendix B, we observe empirically in the results below that the largest gaps in objective value across two consecutive enumerated supports often occur when the consecutive supports belong to different elements of the partition $P_0(\mathcal{D}), P_1(\mathcal{D}), P_2(\mathcal{D}), ..., P_s(\mathcal{D})$. Moreover, as intuitively expected and shown formally in Lemma A.5 of the appendix, increasing $R$ has a positive effect on support recovery. We used $\lambda = 120$ and $(b_x, b_y) = (0.5, 0.5)$.

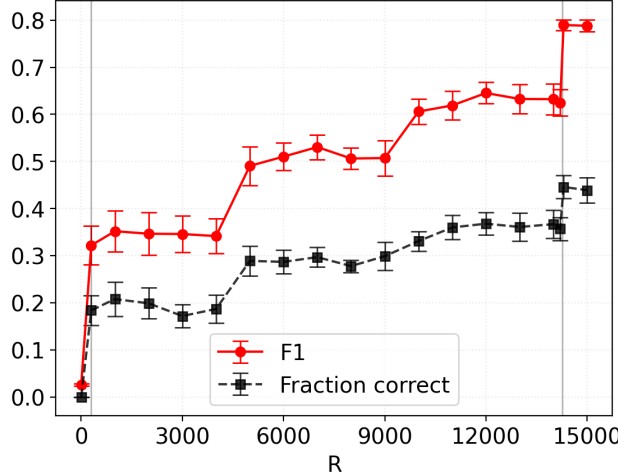

Figure D.3: Numerical experiments for $n = 1{,}000$, $p = 100$, $s = 3$, SNR=5, $\rho = 0.1$, $\epsilon = 1$. Vertical bars denote an objective gap corresponding to an increase in Hamming distance from the optimal support. On the $x$-axis, we vary the value of $R$ and plot the average F1 score and average fraction of correct supports across 10 independent trials for the top-$R$ algorithm. Error bars denote the mean standard error.

**Ablation study of $\lambda$.** We present results below for tuning $\lambda$. We used $R = 2 + (p - s)s$ and $(b_x, b_y) = (0.5, 0.5)$. We observe that choosing a very large $\lambda$ can have a negative effect on support recovery. However, increasing $\lambda$ is beneficial for the runtime of our outer approximation solver. Therefore, we choose a moderate value for $\lambda$ to ensure that the runtime of our method remains comparable to that of MCMC.

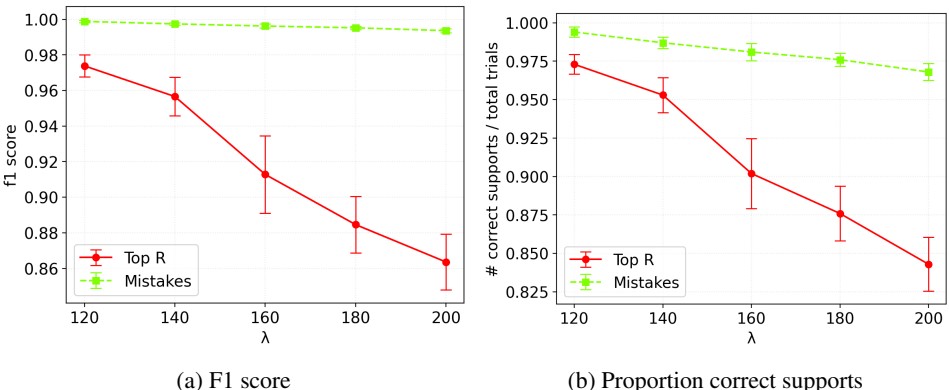

(a) F1 score                    (b) Proportion correct supports

Figure D.4: Numerical experiments for $n = 4{,}000$, $p = 100$, $s = 5$, SNR=5, $\rho = 0.1$, $\epsilon = 1$. On the $x$-axis, we vary the value of $\lambda$, and in Figures D.4a and D.4b and respectively plot the average F1 score and average fraction of correct supports across 10 independent trials for the top-$R$ and mistakes algorithm. Error bars denote the mean standard error.

**Ablation study of $(b_x, b_y)$.** We present results below for tuning $(b_x, b_y)$. For simplicity, we set $b_x = b_y$. We used $R = 2 + (p - s)s$ and $\lambda = 300$. Our results indicate that choosing the clipping constants too small or too large can negatively affect support recovery quality. In practice, these hyperparameters can be tuned via cross-validation.

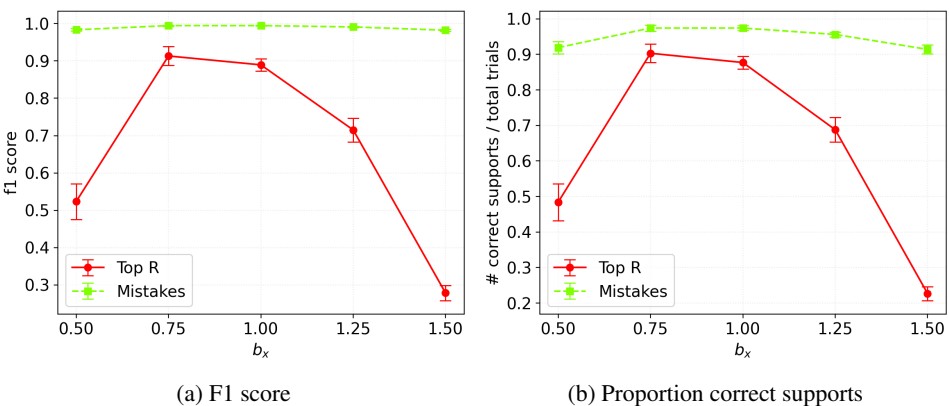

(a) F1 score                    (b) Proportion correct supports

Figure D.5: Numerical experiments for $n = 4{,}000$, $p = 100$, $s = 5$, SNR=5, $\rho = 0.1$, $\epsilon = 1$. On the $x$-axis, we vary the value of $(b_x, b_y)$, and in Figures D.5a and D.5b and respectively plot the average F1 score and average fraction of correct supports across 10 independent trials for the top-$R$ and mistakes algorithm. Error bars denote the mean standard error.

### D.1.3 Support recovery

In this section, we present additional experimental results for support recovery in BSS, for varying values of $s, \rho, \epsilon$, and SNR. As noted in Section 5, the magnitude of the penalty parameter $\lambda$ in Algorithm 2 and the number of MCMC iterations for the algorithm by [29] were chosen such that the average runtimes of the methods across different trials were comparable. That is, our average runtime for each value of $n$ was at most the runtime of the MCMC algorithm.

To demonstrate the power of our MIP-based estimator, we note that for $p = 1000, s = 8$, which is one of the settings that we ran, we have $\binom{1000}{8} \approx 10^{19}$. To put this in perspective, to use the standard exponential mechanism and enumerate all feasible supports, assuming computing each feasible support takes $10^{-6}$ seconds and 16 bits of storage, one would need **764 thousand years and 48 million terabytes of storage**. This shows the usefulness of our MIP-based estimator in practice, enabling us to solve BSS with DP for problem sizes that otherwise would be prohibitive.

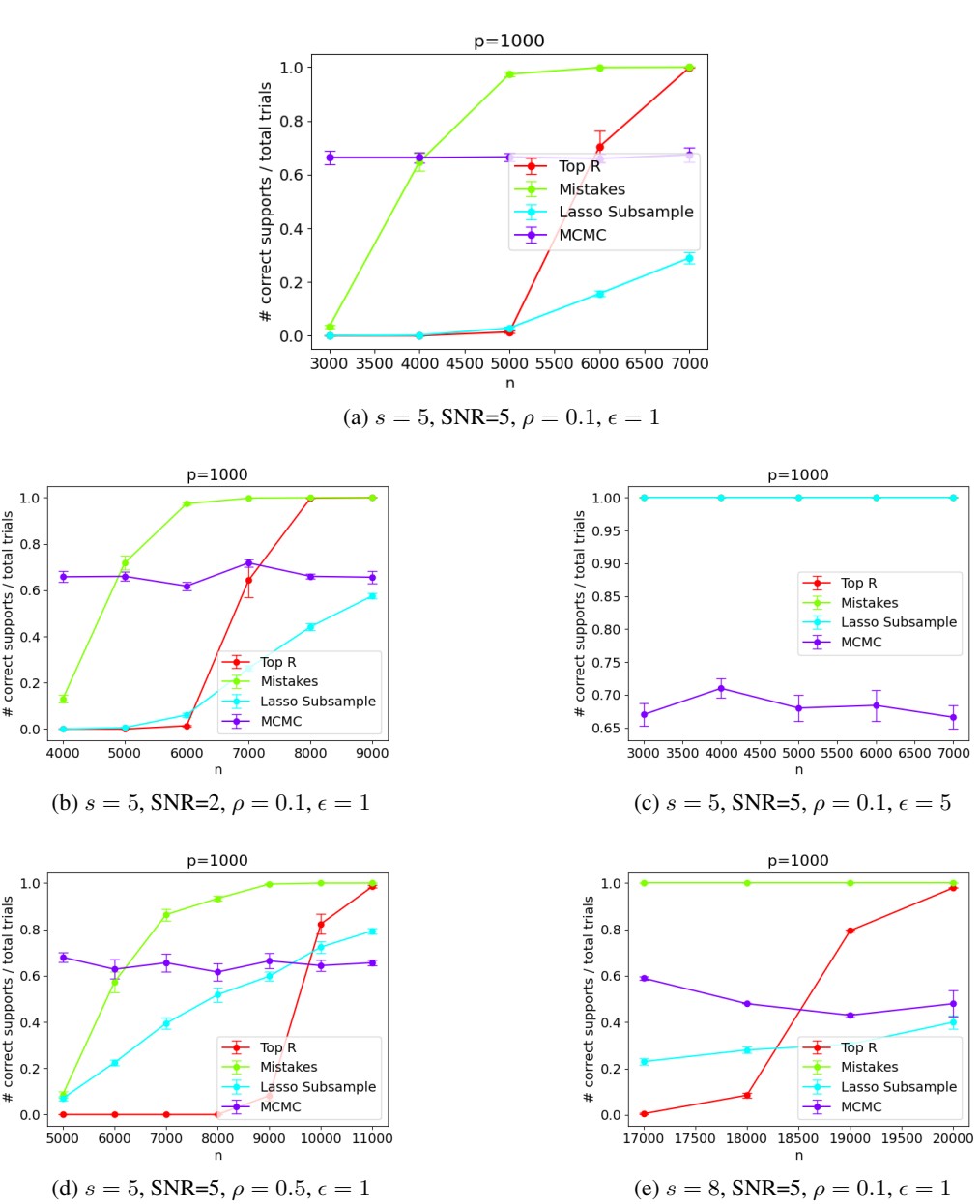

Figure D.6: Numerical experiments for different values of $s, \rho, \epsilon$ and SNR, with $p = 1,000$. The penalty parameter $\lambda$ in Algorithm 2 was set to 250 for figures D.6a-D.6d and 600 for figure D.6e. The number of MCMC iterations was set to 8000 for figures D.6a-D.6d and 10000 for figure D.6e. On the $x$-axis, we vary the value of $n$ and plot the average proportion of draws across 10 independent trials that recovered the right support for each corresponding algorithm. Error bars denote the mean standard error.

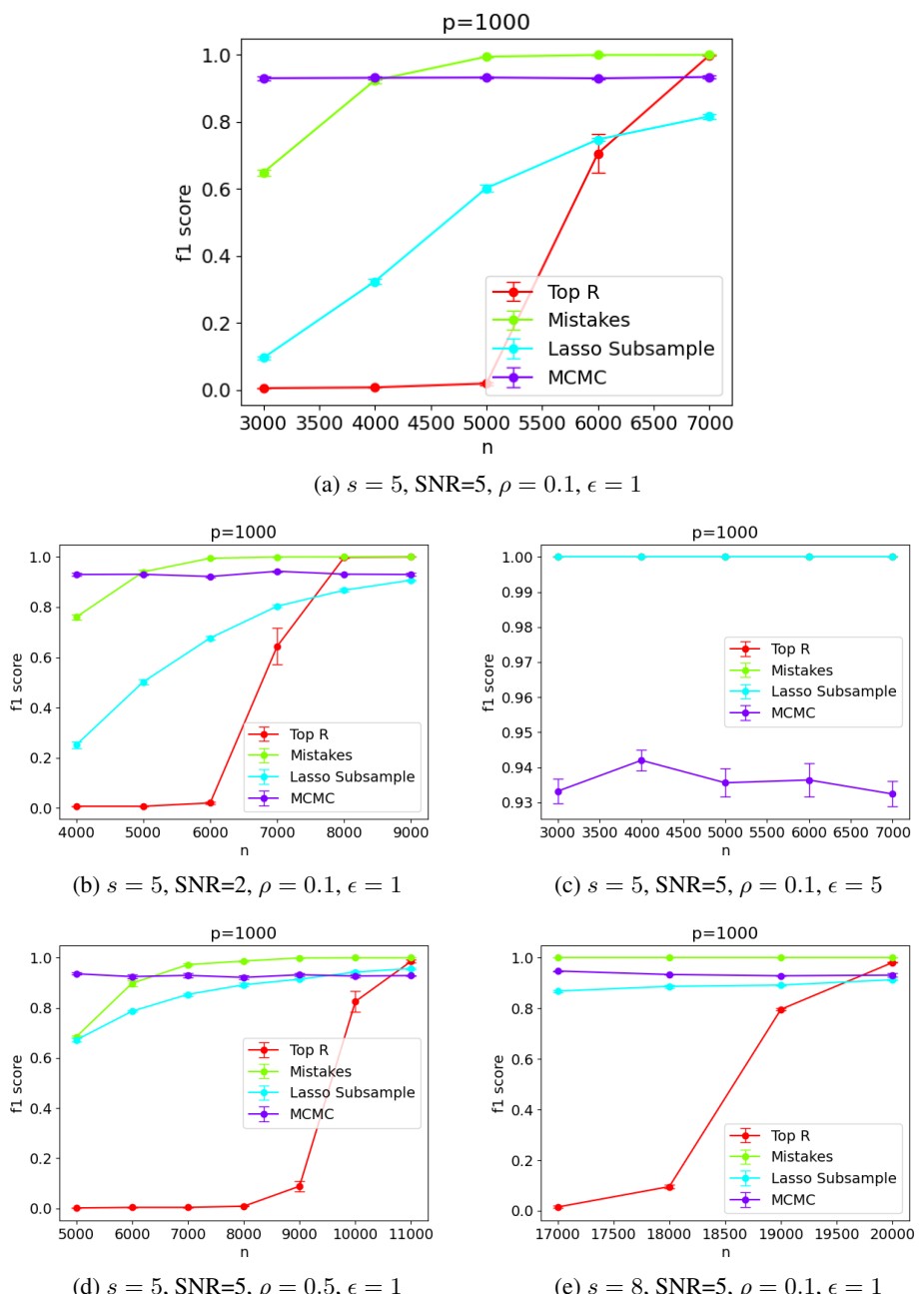

(a) $s = 5$, SNR=5, $\rho = 0.1$, $\epsilon = 1$

(b) $s = 5$, SNR=2, $\rho = 0.1$, $\epsilon = 1$

(c) $s = 5$, SNR=5, $\rho = 0.1$, $\epsilon = 5$

(d) $s = 5$, SNR=5, $\rho = 0.5$, $\epsilon = 1$

(e) $s = 8$, SNR=5, $\rho = 0.1$, $\epsilon = 1$

Figure D.7: Numerical experiments for different values of $s, \rho, \epsilon$ and SNR, with $p = 1,000$. The penalty parameter $\lambda$ in Algorithm 2 was set to $250$ for figures D.7a-D.7d and $600$ for figure D.7e. The number of MCMC iterations was set to $8000$ for figures D.7a-D.7d and $10000$ for figure D.7e. On the $x$-axis, we vary the value of $n$ and plot the average F1 score across 10 independent trials for each corresponding algorithm. Error bars denote the mean standard error.

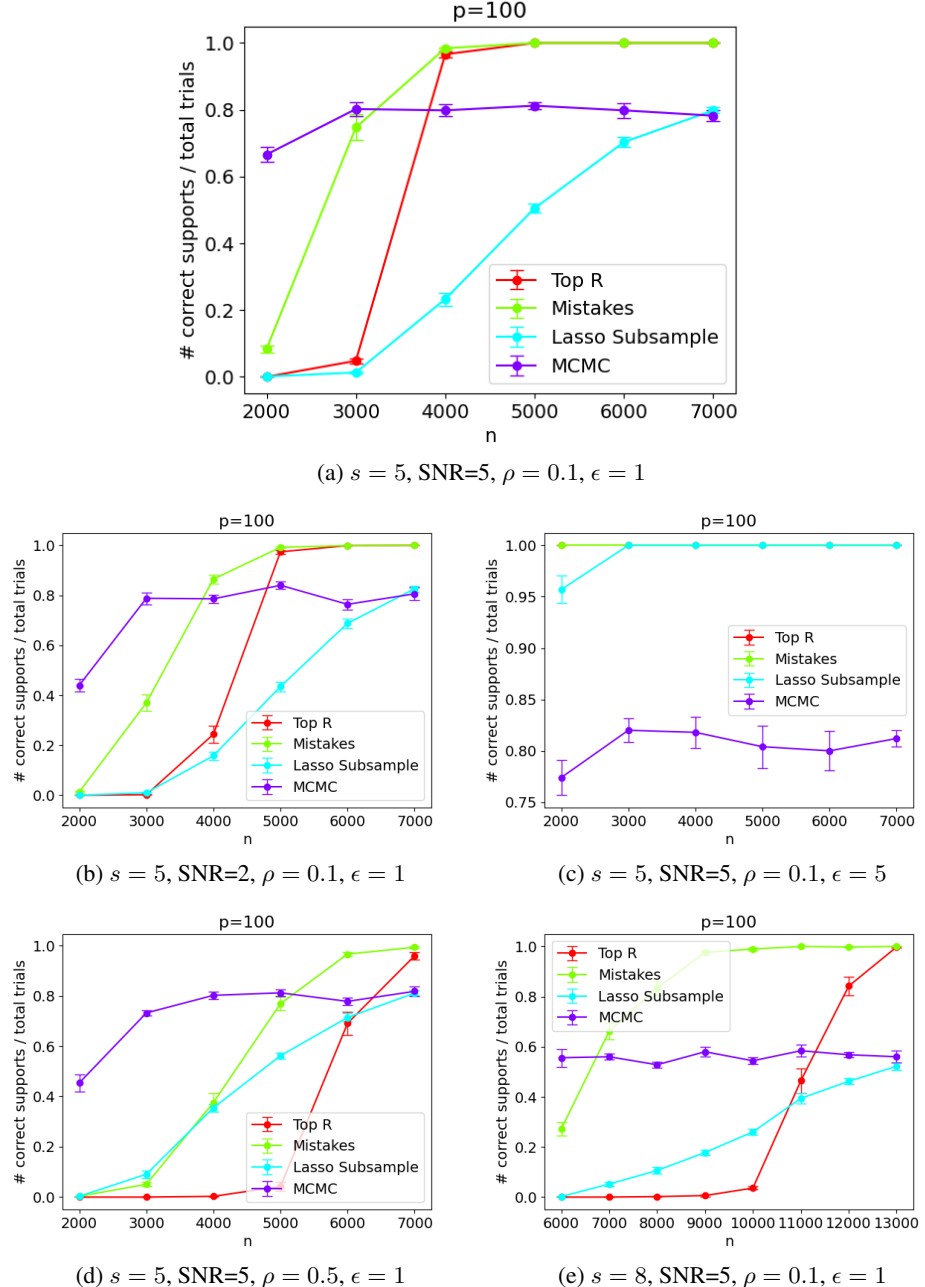

(a) $s = 5$, SNR=5, $\rho = 0.1$, $\epsilon = 1$

(b) $s = 5$, SNR=2, $\rho = 0.1$, $\epsilon = 1$

(c) $s = 5$, SNR=5, $\rho = 0.1$, $\epsilon = 5$

(d) $s = 5$, SNR=5, $\rho = 0.5$, $\epsilon = 1$

(e) $s = 8$, SNR=5, $\rho = 0.1$, $\epsilon = 1$

Figure D.8: Numerical experiments for different values of $s, \rho, \epsilon$ and SNR, with $p = 100$. The penalty parameter $\lambda$ in Algorithm 2 was set to $120$ for figures D.8a-D.8d and $350$ for figure D.8e. The number of MCMC iterations was set to $1000$ for figures D.8a-D.6e. On the $x$-axis, we vary the value of $n$ and plot the average proportion of draws across 10 independent trials that recovered the right support for each corresponding algorithm. Error bars denote the mean standard error.

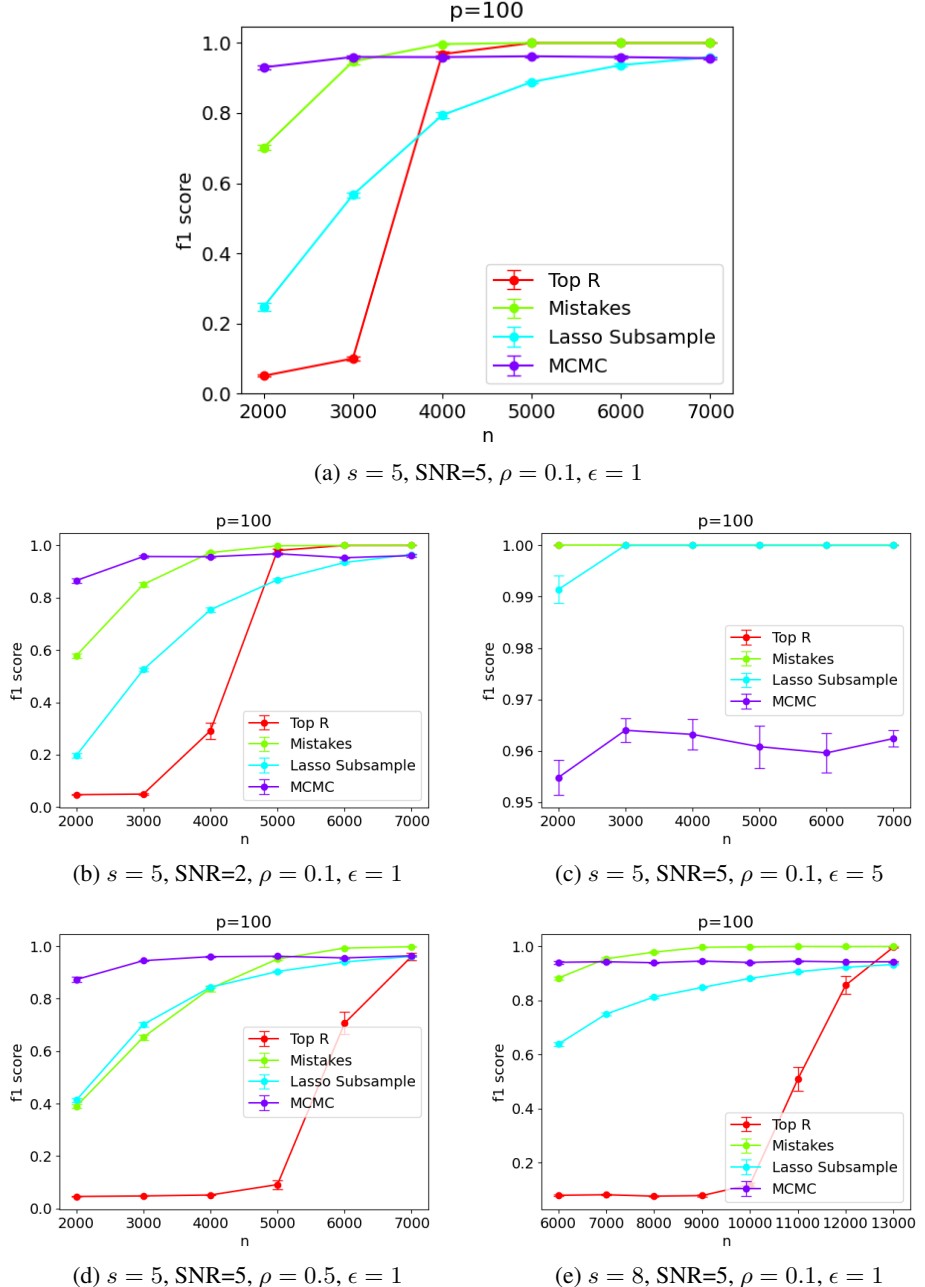

Figure D.9: Numerical experiments for different values of $s, \rho, \epsilon$ and SNR, with $p = 100$. The penalty parameter $\lambda$ in Algorithm 2 was set to 120 for figures D.9a-D.9d and 350 for figure D.9e. The number of MCMC iterations was set to 1000 for figures D.9a-D.7e. On the $x$-axis, we vary the value of $n$ and plot the average F1 score across 10 independent trials for each corresponding algorithm. Error bars denote the mean standard error.

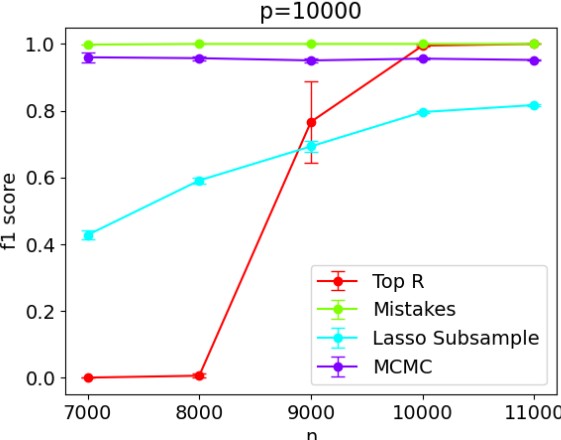

Figure D.10: Numerical experiments for $p = 10,000$, with $s = 5$, SNR=5, $\rho = 0.1$, and $\epsilon = 1$. The penalty parameter $\lambda$ in Algorithm 2 was set to 600, and the number of MCMC iterations was set to $100,000$. On the $x$-axis, we vary the value of $n$ and plot the average F1 score across 10 independent trials for each corresponding algorithm. Error bars denote the mean standard error.

## D.2 Hinge Loss

In this section, we present our experimental results in the setting of sparse classification, presented in Problem C.1.

To generate our data, we first generate $z_i = \boldsymbol{x}_i^T \boldsymbol{\beta}^* + \epsilon_i$ for $i \in [n]$, where $\boldsymbol{x}_1, \cdots, \boldsymbol{x}_n \overset{\text{iid}}{\sim} \mathcal{N}(\boldsymbol{0}, \boldsymbol{\Sigma}) \in \mathbb{R}^p$ and the independent noise follows $\boldsymbol{\epsilon} \sim \mathcal{N}(\boldsymbol{0}, \sigma^2 \boldsymbol{I}_n)$ where $\boldsymbol{I}_n$ is the identity matrix of size $n$. We then draw $u_i \sim \text{Uniform}[0,1]$ for $i \in [n]$, and we set

$$y_i = \begin{cases} 1 & \text{if } u_i > \sigma(z_i) \\ -1 & \text{otherwise} \end{cases} \quad \text{where } \sigma(z) = \frac{1}{1 + e^{-z}}.$$

Moreover, for $i, j \in [p]$, we set $\Sigma_{i,j} = \rho^{|i-j|}$ and set nonzero coordinates of $\boldsymbol{\beta}^*$ to take value $1/\sqrt{s}$ at indices $\{1, 3, \cdots, 2s - 1\}$. We define the Signal to Noise Ratio as SNR $= \|\boldsymbol{X}\boldsymbol{\beta}^*\|_2^2 / \|\boldsymbol{\epsilon}\|_2^2$.

As with the BSS results, our methods show favorable empirical support recovery in both low and high-dimensional settings, with our mistakes method outperforming our top-$R$ method.

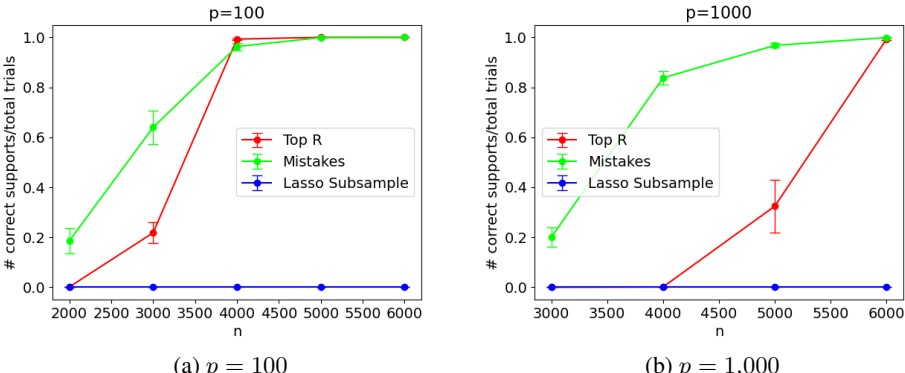

(a) $p = 100$                    (b) $p = 1,000$

Figure D.11: Numerical experiments for $p = 100$ and $p = 1,000$, with $s = 5$, SNR=5, $\rho = 0.1$, and $\epsilon = 1$. The penalty parameter $\lambda$ in Algorithm 2 was set to 90 and 100 for figures D.11a and D.11b, respectively. On the $x$-axis, we vary the value of $n$ and plot the average proportion of draws across 10 independent trials that recovered the right support for each corresponding algorithm. Error bars denote the mean standard error.

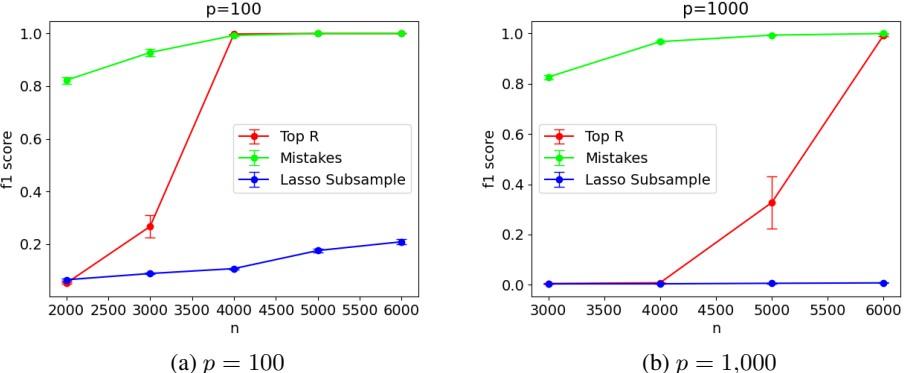

(a) $p = 100$        (b) $p = 1,000$

Figure D.12: Numerical experiments for $p = 100$ and $p = 1,000$, with $s = 5$, SNR=5, $\rho = 0.1$, and $\epsilon = 1$. The penalty parameter $\lambda$ in Algorithm 2 was set to 90 and 100 for figures D.12a and D.12b, respectively. On the $x$-axis, we vary the value of $n$ and plot the average F1 score across 10 independent trials for each corresponding algorithm. Error bars denote the mean standard error.

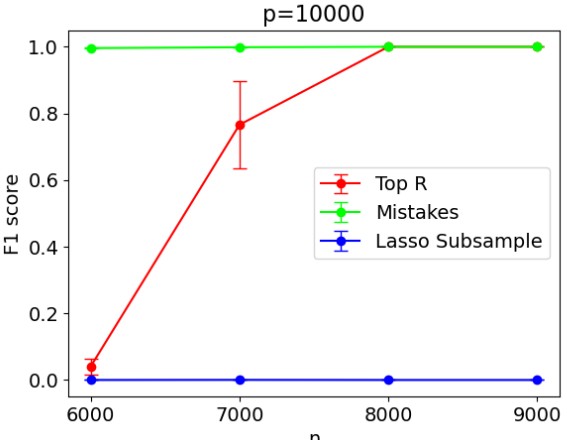

Figure D.13: Numerical experiments for $p = 10,000$, with $s = 5$, SNR=5, $\rho = 0.1$, and $\epsilon = 1$. The penalty parameter $\lambda$ in Algorithm 2 was set to 170. On the $x$-axis, we vary the value of $n$ and plot the average F1 score across 10 independent trials for each corresponding algorithm. Error bars denote the mean standard error.

