# OpenReview forum: "Differentially Private High-dimensional Variable Selection via Integer Programming"
_NeurIPS.cc/2025/Conference — NeurIPS 2025 poster_

### Official Review · Reviewer_PKV9 · 2025-06-24

**Clarity:** 3
**Significance:** 2
**Originality:** 3
**Rating:** 3
**Confidence:** 4

**Summary:**

This paper focuses on the sparse variable selection in the context of Differential Privacy (DP), which aims to improve interpretability and generalization in high-dimensional learning while protecting the confidential and personal information. This paper introduces two new differentially private estimators for sparse variable selection based on modern MIP techniques. Their framework is broadly applicable, and they provide theoretical support recovery guarantees in the case of sparse regression. Finally, extensive numerical experiments, using both least squares and hinge loss, demonstrate the effectiveness and scalability of their methods.

**Questions:**

Please see weakness above.
A minor comment is: the notation $\rho$  is used to represent different concepts.

**Ethical Concerns:**

["NO or VERY MINOR ethics concerns only"]

**Final Justification:**

While I believe this paper brings forth an interesting idea, there exist technical limitations in their methodology. Thus, I maintain my score.

**Limitations:**

Some results of this paper are limited to the sparse regression problem, as detailed in the weakness. It would be beneficial to extend these results to broader settings and include numerical results for a wider class of loss functions.

**Quality:**

2

**Strengths And Weaknesses:**

This paper is clearly written and well-structured. This paper is the first to apply MIP techniques for sparse variable selection in the context of DP. The work addresses an important problem in the field of variable selection.

However, the computational results could be improved. Specifically, the authors do not discuss why the Top-R method performs better as the sample size n increases. Throughout the computational experiments, the sparsity parameter s is set to 5 or 8, much smaller compared to the number of features. This may not be representative. It is necessary to try larger values of s to evaluate the methods.

While the two DP algorithms are general, the support recovery results are limited to the sparse regression setting. In addition, they only provide MIP formulations and algorithms for least squares and hinge loss objectives. It remains unclear how MIP techniques extend to general loss function.

---

> ### Author Rebuttal · Authors · 2025-07-31
>
> Dear reviewer PKV9,
>
> We sincerely appreciate your feedback on our paper. We would like to address your concerns below.
>
> **Effect of sample size on Top-R performance**
>
> Thank you for your comment! The performance improves with increasing $n$ due to the decrease in the threshold needed for the identifiability margin $\tau$ to have enough separation between the true support and other supports (Theorem 3 in our paper). We will make sure to update this in our manuscript.
>
> **Empirical performance with larger s**
>
> In order to evaluate the performance of our methods for larger values of $s$, we ran experiments with $p=100, s=20, \epsilon = 1, SNR= 5,$ and $\rho = 0.1$. For each value of $n$, we ran 10 independent trials, where, for each trial, we drew 100 times from the distribution corresponding to each algorithm, and for MCMC, we used 100 independent Markov chains for each trial. Our choice of $\lambda=80,000$ and MCMC iterations = 1000 resulted in our mistakes method having a considerably smaller average runtime than the MCMC method.
>
> We present our results below. As expected, we observed that for $n$ large enough, our mistakes method outperformed both the Samp-Agg and MCMC methods.
>
> Average # correct supports / number of trials:
>
> |               | $n=120,000$ | $n=130,000$ | $n=140,000$ | $n=150,000$ |
> |---------------|-----------|-----------|-----------|-----------|
> | mistakes  | 0.31      | 0.49      | 0.88      | 0.97      |
> | Samp-Agg | 0.71      | 0.79      | 0.82      | 0.83      |
> | MCMC      | 0.13      | 0.12      | 0.06      | 0.10      |
>
> Average F1 scores:
>
> |               | $n=120,000$ | $n=130,000$ | $n=140,000$ | $n=150,000$ |
> |---------------|-----------|-----------|-----------|-----------|
> | mistakes  | 0.9535    | 0.9735    | 0.994     | 0.998     |
> | Samp-Agg  | 0.985     | 0.9895    | 0.99      | 0.992     |
> | MCMC      | 0.936     | 0.936     | 0.923     | 0.94      |
>
> **MIP extension to general loss functions**
>
> A key benefit of our MIP approach is its generalizability across different loss functions. Specifically, the only property that we need to have for the loss function $\rho$ is that it is a convex function of $\beta$. As long as the loss function is convex, our outer approximation approach can be applied to any general loss function. For example, our method works with Huber or quantile loss as well, and the MIP algorithm would remain unchanged.
>
> **Support recovery results are limited to the sparse regression setting**
>
> We acknowledge that our theoretical support recovery guarantees are currently limited to the sparse regression setting. However, extending these results rigorously to more general loss functions would necessitate a fundamentally different analytical framework and set of assumptions than that used in the least squares setting [1]. Such an extension would be substantial enough to warrant dedicated treatment in future work. Additionally, we emphasize that, to our knowledge, the existing theoretical literature on differentially private sparse variable selection explicitly focuses on linear regression. Therefore, restricting our theoretical support recovery analysis to the sparse regression context aligns naturally with the current state of research and represents a meaningful contribution within this established framework.
>
> **Comment regarding notational conflict**
>
> Thank you for noticing our notational conflict with the use of $\rho$. We will make sure to adjust this in our manuscript.
>
> References:
>
> [1] Chen, L. Y., & Lee, S. (2023). Sparse quantile regression. Journal of Econometrics, 235(2), 2195–2217.

---

> > ### Comment · Reviewer_PKV9 · 2025-08-04
> >
> > Thank you for your response. It has addressed several of my questions. I trust the reviewers will include any clarifying details in their final revision.

---

> > > ### Author Response · Authors · 2025-08-04
> > >
> > > Dear Reviewer PKV9,
> > >
> > > Thank you very much for your response to our rebuttal. As you noted, we will make sure to incorporate all relevant clarifications in our final revision.
> > >
> > > Best regards,
> > >
> > > Authors

---

### Official Review · Reviewer_vS6c · 2025-06-25

**Clarity:** 3
**Significance:** 3
**Originality:** 4
**Rating:** 5
**Confidence:** 3

**Summary:**

The paper introduces recent developments in Mixed Integer Programming (MIP) for the Best Subset Selection (BSS) problem to the differential privacy community. It presents two selection procedures: Top-R, which is based on the exponential mechanism and either selects one of the top R subsets or samples a subset uniformly at random; and the "Mistakes" method, which applies the exponential mechanism based on the distance (the number of mistakes) from the optimal solution. To run the Top-R and Mistakes mechanisms, one needs to get the top R subsets or the best subsets with a fixed number of mistakes, which the authors do by sequentially solving modified MIP problems. This approach allows private BSS problems to scale up to $10^4$ parameters.

**Questions:**

You state that Algorithm 1 satisfies $(\epsilon, 0)$-DP, but according to Theorem 1, it actually satisfies $(\epsilon', 0)$-DP unless $T = \infty$. Is this a typo?

Are there similar modifications of the exponential mechanism in the literature? If so, it would be advisable to mention them in the related works section.

**Ethical Concerns:**

["NO or VERY MINOR ethics concerns only"]

**Final Justification:**

The authors developed an efficient algorithm for differentially private Best Subset Selection by modifying the exponential mechanism to sample not from the full exponential space of all possible subsets, but only from the best ones. I consider this paper an important contribution: in numerical experiments, the Mistakes and Top-R methods significantly outperform existing alternatives. Furthermore, the paper provides a non-trivial theoretical analysis of the utility guarantees of the proposed methods in terms of support recovery (Theorems 3 and 4). For these reasons, I support the acceptance of the paper.

During the rebuttal, the authors addressed my concerns about the theoretical guarantees and the presentation of the paper.

However, the extra assumptions required for the Mistakes methods remain a major weakness.

**Limitations:**

yes

**Paper Formatting Concerns:**

The formulas in the Appendix need to be reformatted to fit within the page margins.

**Quality:**

4

**Strengths And Weaknesses:**

The authors developed an efficient algorithm for differentially private Best Subset Selection by modifying the exponential mechanism to sample not from the full exponential space of all possible subsets, but only from the best ones. To my knowledge, this adaptation of the exponential mechanism is original, though similar ideas may potentially exist. I consider this paper an important contribution: in numerical experiments, the Mistakes and Top-R methods significantly outperform existing alternatives. Furthermore, the paper provides a non-trivial theoretical analysis of the utility guarantees of the proposed methods in terms of support recovery (Theorems 3 and 4). For these reasons, I support the acceptance of the paper.

Weaknesses:

1) The theoretical guarantees for support recovery appear to be weaker than those provided by the MCMC method [22].

2) The privacy analysis of the Mistakes method (Theorem 2) relies on extra assumptions that may not hold in practice, potentially limiting the method’s applicability.

3) The presentation of the paper could also be improved. The conclusion section is unnecessarily short, and the font sizes in the plots are too small, making them difficult to read. I would also suggest renaming the "Mistakes" method, as the current name might be misleading. Furthermore, the formulas in the Appendix need to be reformatted to fit within the page margins.

---

> ### Author Rebuttal · Authors · 2025-07-31
>
> Dear reviewer vS6c,
>
> We would like to sincerely thank you for your positive feedback on our paper. We would like to address your concerns below.
>
> **Theoretical guarantees for support recovery**
>
> We would like to clarify that our two methods match the best theoretical guarantees in the literature in complementary privacy regimes, as discussed in the contributions section of our paper and in section 3. Namely, our top-R method matches with the non-private minimax-optimal signal threshold for support recovery in the low privacy regime, and our mistakes method matches the required signal threshold in [1] in the high-privacy regime.
>
> **Privacy of mistakes method**
>
> We would like to note that while the privacy of our mistakes method relies on an additional assumption that occurs with high probability (as shown in section A.5 in the appendix), providing a privacy guarantee with additional assumptions is not uncommon in the literature. For example, the privacy guarantees of [1], which is the closest competitor to our method, depends on assumptions that hold with high probability. More specifically, the privacy proof of the Markov Chain Monte Carlo (MCMC) algorithm in [1] relies on the assumption that the mixing of the Markov Chain used for sampling with its stationary distribution has happened. However, this mixing can only be guaranteed with high probability, and under additional assumptions on the underlying model—see Theorem 4.3 of [1] for more details. In contrast, our top-R method is always private (assuming $b_x,b_y$ are finite), and our mistakes method is private under assumptions that are similar to the ones in Theorem 4.3 of [1].
>
> As another example, [2] uses the stability of Lasso, to present a DP method for support recovery in sparse linear regression. However, the stability of Lasso only holds under certain assumptions on the data, such as the boundedness of the noise and restricted strong convexity. Such assumptions might only hold with high probability in practice, resulting in privacy guarantees that hold with high probability. For more details, we refer to Theorem 8 of [2].
>
> **Presentation and typos**
>
> Thank you very much for correcting our typo in regards to the privacy of our top-R method. As you correctly point out, top-R is $(\epsilon’,0)$-DP unless $T=\infty$. Thank you also for noting how our paper presentation could be improved. We will make sure to edit the formulas in the appendix to fit within the page margins, grow the font sizes in the plots, and expand our conclusion section further.
>
> **Similar modifications of the exponential mechanism**
>
> In regards to your last question, there are no similar modifications to the exponential mechanism in the literature to our knowledge.
>
> References:
>
> [1] Roy, S., Wang, Z., & Tewari, A. (2024). On the computational complexity of private high-dimensional model selection via the exponential mechanism. In Advances in Neural Information Processing Systems (NeurIPS).
>
> [2] Thakurta, A. G., & Smith, A. (2013). Differentially private feature selection via stability arguments, and the robustness of the lasso. In Proceedings of the Conference on Learning Theory (COLT), PMLR.

---

> > ### Comment · Reviewer_vS6c · 2025-08-04
> >
> > Thank you for answering my questions, I will keep supporting the paper's acceptance!
> >
> > I however believe that there are papers that truncate exponential mechanism, for example those two seem to do so
> >
> > **Truncated Exponential Mechanism**
> >
> > Carvalho, Ricardo Silva, et al. "TEM: High utility metric differential privacy on text." Proceedings of the 2023 SIAM International Conference on Data Mining (SDM). Society for Industrial and Applied Mathematics, 2023.
> >
> > **Restricted Exponential Mechanism**
> >
> > Brown, Gavin, et al. "Covariance-aware private mean estimation without private covariance estimation." Advances in neural information processing systems 34 (2021): 7950-7964.

---

> > > ### Author Response · Authors · 2025-08-04
> > >
> > > Dear Reviewer vS6c,
> > >
> > > Thank you very much for your continued support of our paper and for pointing us to two papers in the DP literature that truncate the exponential mechanism. We will incorporate these references in our related work section.
> > >
> > > Best regards,
> > >
> > > Authors

---

### Official Review · Reviewer_JE2d · 2025-06-26

**Clarity:** 4
**Significance:** 3
**Originality:** 4
**Rating:** 5
**Confidence:** 3

**Summary:**

The paper focuses on the problem of sparse variable selection with differential privacy. Here we have some per-example loss (such as least squares or the hinge loss) on a model $\beta$, a dataset $X, y$, and our goal is to find a model $\beta$ that minimizes the empirical loss on $X$ subject to the model being chosen with differential privacy, and $\beta$ having at most $s$ non-zero coordinates. Variable selection focuses just on selecting the right $s$ coordinates; in particular, the success of a variable selection algorithm is the fraction of non-zero coordinates in $\beta^*$, the optimal model, that are also chosen by the algorithm.

The authors consider an exponential mechanism-based approach to this problem. In their first variant, the loss for choosing a certain set of $s$ coordinates is the minimum loss achieved by any model only supported on those coordinates. They are able to show a bound on the sensitivity of the exponential mechanism under mild assumptions (that can be enforced via clipping), but since there are $\binom{p}{s}$ different sets of coordinates to choose, running this exponential mechanism is infeasible. They instead propose the following modification: rather than compute the loss for all sets, compute it for the best r sets. For all other sets, set their loss to the r-th best set's loss. They show this is also private, and in fact gets better privacy than the "default" value of $r = \binom{p}{s}$ improving as r decreases. They also show that the top R supports can be computed by solving a series of mixed integer programs which are efficient in practice for p ~= 10000. This eliminates the need to enumerate all solutions. The second variant the authors propose is to set the loss in the exponential mechanism to be the number of wrong coordinates in the set. This is easier to sample from but requires a separation between the loss of the first and second for privacy.

Theoretically, the authors show "top-r" recovers the best selection with high probability when the smallest non-zero coordinate of the optimal model is $\Omega(\sqrt{\max\{1, s^2 / \epsilon\} log p / n})$ for a moderate value of r. The non-private requirement is $\Omega(\sqrt{\log p / n})$, so top-R matches the non-private rate up to constants when $\epsilon \rightarrow \infty$ or when the number of coordinates to pick is small. Mistakes requires the non-zero coordinates to be $\Omega(\sqrt{(s / \epsilon) \cdot \log p / n})$, i.e. again is optimal when $\epsilon$ is sufficiently large but requires the additional separation condition for privacy. Empirically, the authors compare top-r and mistakes to an MCMC-based approach of the previous work of Roy et al., and also the Samp-Agg algorithm of Smith et al. Mistakes recovers almost all coordinates in most settings, and top-R outperforms Samp-Agg and outperforms the MCMC based approach when the dataset size is large enough.

**Questions:**

-Can the separation between the loss of the first and second-best solution be enforced in any clipping-like way? As discussed before, requiring a high-probability condition to hold for the DP guarantee of mistakes is somewhat problematic.
-If we were to e.g. run objective perturbation or another DP-ERM algorithm over the coordinates selected by top-r or mistakes using the same overall privacy budget as Samp-Agg, would top-r and mistakes still outperform Samp-Agg?

**Ethical Concerns:**

["NO or VERY MINOR ethics concerns only"]

**Final Justification:**

The rebuttal has addressed my concerns. It would be nice if mistakes could be made unconditionally private, or if the authors could include suggestions for dealing with the potential for it to not be private, but the good performance of top-K and the fact that the previous work also included such a conditional privacy guarantee makes this a more minor weakness than I had originally thought. The new experiments make the empirical improvement of the authors' method more convincing and addressed my main concern about the paper.

**Limitations:**

Yes.

**Quality:**

3

**Strengths And Weaknesses:**

Strengths:
* Best coordinate selection is an interesting and well-studied problem.
* A number of theoretical insights in combination are needed for the results. e.g. top-R requires the combination of a sensitivity bound which is non-trivial, a cleverly-designed MIP to isolate the top-R sets without brute-force enumeration, and the authors also give a privacy proof which shows that imposing a ceiling on the loss function in the exponential mechanism improves its utility.
* The algorithms are tight in that match the non-private best rates of recoery as $\epsilon \rightarrow \infty$.
* The presentation makes the flow of ideas appear to be natural which makes for an easy read.

Weaknesses:
* The mistakes algorithm's privacy guarantee relies on a separation between the loss of the best and second-best datasets, i.e. the privacy guarantee doesn't hold for all pairs of datasets. While the authors argue under certain sampling assumptions for the datasets, this separation holds with high probability, DP is inherently a worst-case statement about all pairs of distributions so "average-case" DP definitions are generally problematic in nuanced ways (see e.g. https://differentialprivacy.org/average-case-dp/ for more discussion) and such definitions shouldn't be promoted in the literature. So the mistakes algorithm is probably not desirable (except maybe as a theoretical ideal) unless e.g. the separation condition can be forced using clipping-style arguments.
* The empirical comparison studies the chance of recovering the right set of coordinates, but the final downstream objective is actually to minimize the model loss and not just find the best coordinates. Samp-Agg in particular does minimize the final loss, so a comparison to Samp-Agg is a bit unfair because (i) Samp-Agg is not optimizing for the highest % of correct coordinates, and (ii) Samp-Agg is spending part of its privacy budget on the model optimization, whereas top-R and mistakes do not spend any privacy budget on this aspect of the problem.
* The combination of the two above points makes the empirical comparisons a bit unconvincing, since (i) mistakes has a problematic privacy guarantee, (ii) top-R does not uniformly beat MCMC, and (iii) top-R beats Samp-Agg, but that comparison is unfair.

---

> ### Author Rebuttal · Authors · 2025-07-31
>
> Dear reviewer JE2d,
>
> We sincerely appreciate your effort in providing detailed feedback on our paper. We address your comments below.
>
> **Privacy of mistakes method**
>
> We would like to note that while the privacy of our mistakes method relies on an additional assumption that occurs with high probability (as shown in section A.5 in the appendix), providing a privacy guarantee with additional assumptions is not uncommon in the literature. For example, the privacy guarantees of [1], which is the closest competitor to our method, depends on assumptions that hold with high probability. More specifically, the privacy proof of the Markov Chain Monte Carlo (MCMC) algorithm in [1] relies on the assumption that the mixing of the Markov Chain used for sampling with its stationary distribution has happened. However, this mixing can only be guaranteed with high probability, and under additional assumptions on the underlying model—see Theorem 4.3 of [1] for more details. In contrast, our top-R method is always private (assuming $b_x,b_y$ are finite), and our mistakes method is private under assumptions that are similar to the ones in Theorem 4.3 of [1].
>
> As another example, [2] uses the stability of Lasso, to present a DP method for support recovery in sparse linear regression. However, the stability of Lasso only holds under certain assumptions on the data, such as the boundedness of the noise and restricted strong convexity. Such assumptions might only hold with high probability in practice, resulting in privacy guarantees that hold with high probability. For more details, we refer to Theorem 8 of [2].
>
>
>
> **Comparison with Samp-Agg**
>
> We would first like to clarify that in our experiments, we do not compare with Algorithm 3 in [3], which allocates half the privacy budget on feature selection and half on model optimization, but rather, we are comparing with Algorithm 5 which exclusively is aimed at feature selection for the same problem and is itself (ε,0)-DP.
>
> Nevertheless, as you suggested, in an effort to compare prediction accuracy across methods, we did a 70/30 random train test/split and implemented Algorithm 2 in [3] on the train data, wherein we use half the privacy budget (i.e., $\epsilon/2$) for variable selection using the top-R, mistakes, Samp-Agg, or MCMC methods, and the remaining half of the budget for model optimization using objective perturbation (Algorithm 1 in [3]) to obtain the regression coefficients under the privacy budget ($\beta_\text{priv}$). We ran experiments with $p=100, s=5, \epsilon = 2, SNR= 5, \rho = 0.1$, and ran 10 independent trials for each value of $n$, where, for each trial, we drew 50 times from the distribution corresponding to each algorithm, and for MCMC, we used 50 independent Markov chains for each trial. After running objective perturbation over the supports, we thus had 500 distinct coefficient vectors for each method and for each $n$, and we computed the average MSE on the test data for every method. Our choices of $\lambda=120$ and MCMC iterations = 1000 were chosen to make the runtimes comparable, as done in our paper. We summarize our results below, showing that our top-R and mistakes methods outperform the competitor algorithms in terms of prediction accuracy for $n$ large enough.
>
> Average test MSE:
>
> |                             | $n=3000$ | $n=4000$ | $n=5000$ | $n=6000$ |
> |-----------------------------|-----------|-----------|-----------|-----------|
> | top-R                       | 1.894    |  0.650    | 0.290    | 0.276    |
> | mistakes                    | 1.064    | 0.408    | 0.300    | 0.275    |
> | Samp-Agg                    | 1.490   | 0.980    | 0.677    | 0.480    |
> | MCMC                        | 0.532    | 0.386    | 0.329   | 0.317   |
>
> Furthermore, we ran experiments to evaluate the utility loss, or gap between the objective at $\beta_\text{priv}$ and the objective at $\hat{\beta}$ (the optimal BSS solution), with the same parameters as in the experiments above and we present our results below. As with the prediction accuracy, our top-R and mistakes methods outperform the competitor algorithms in terms of utility loss for $n$ large enough.
>
> Average utility loss:
>
> |                             | $n=3000$ | $n=4000$ | $n=5000$ | $n=6000$ |
> |-----------------------------|-----------|-----------|-----------|-----------|
> | top-R                       | 0.192   |  0.020    | 0.005   | 0.003    |
> | mistakes                    |0.061   | 0.016   | 0.005    | 0.003   |
> | Samp-Agg                    | 0.082   | 0.037    | 0.018    | 0.011    |
> | MCMC                        | 0.017    | 0.013    | 0.010   | 0.008   |
>
> References:
>
> [1] Roy, S., Wang, Z., & Tewari, A. (2024). On the computational complexity of private high-dimensional model selection via the exponential mechanism. In Advances in Neural Information Processing Systems (NeurIPS).
>
> [2] Thakurta, A. G., & Smith, A. (2013). Differentially private feature selection via stability arguments, and the robustness of the lasso. In Proceedings of the Conference on Learning Theory (COLT), PMLR.
>
> [3] Kifer, D., Smith, A., & Thakurta, A. (2012). Private convex empirical risk minimization and high-dimensional regression. In Proceedings of the Conference on Learning Theory (COLT), JMLR Workshop and Conference Proceedings.

---

> ### Comment · Reviewer_JE2d · 2025-08-04
>
> Thanks for your response. Overall I feel more positive about the paper after reading the authors' response and will raise my score accordingly.
>
> Re: Privacy of mistakes method - I think [2] is a bit outdated, but I appreciate that [1] is the paper the authors are directly comparing against, and to that end one can reframe this point as "the unconditional privacy of top-K is an improvement over past work" rather than "the conditional privacy of mistakes is a limitation". I'm willing to concede that this alone should not give a negative rating to the paper.
>
> Re: The new experiments. Thanks for adding these and for the clarification. I think these results address my concerns, they are much more compelling and a great addition to the paper.

---

> > ### Author Response · Authors · 2025-08-04
> >
> > Dear Reviewer JE2d,
> >
> > Thank you very much for your thoughtful response and feedback and for raising your score. Your suggestions have certainly helped strengthen our paper.
> >
> > Best regards,
> >
> > Authors

---

### Official Review · Reviewer_fXeC · 2025-07-06

**Clarity:** 3
**Significance:** 4
**Originality:** 3
**Rating:** 4
**Confidence:** 2

**Summary:**

Sparse variable selection is important for high-dimensional data, it can improve the interpretability and reduce the overfitting of the models. This paper proposes two new and scalable differentiable privacy estimators for sparse variable selection, “top-R” and “mistake”. And both of them employ the idea of mixed integer programming to generate the candidate supports, avoiding the exhaustive search in the exponential mechanism. They also prove that these two estimators satisfy pure (ε,0)-DP. And the best subset selection matches the non-private minimax rate of top-R in the low privacy regime and those of mistakes in the high privacy regime. Experiments on least squares regression and hinge classification prove that the proposed method is able to recover the correct support better than SOTA, and is still computationally feasible.

**Questions:**

How to choose the parameters R and λ? An ablation study would be useful.
In practice, how should one set the clipping bounds bx and by to balance bias and privacy?

**Ethical Concerns:**

["NO or VERY MINOR ethics concerns only"]

**Limitations:**

Yes.

**Paper Formatting Concerns:**

None.

**Quality:**

3

**Strengths And Weaknesses:**

Strength:
1) This paper gave us an insight into using MIP solvers to approximate the exponential mechanism and solve the problem that effectively solves combinatorial optimization problems under the strong guarantee of pure-DP.
2) The author provided solid theoretical proof. They provide a detailed proof for pure (ε,0)-DP and analyze the support recovery for BSS.
3) They gave an empirical comparison with state-of-the-art methods, showing great results.

Weakness:
1) The proposed methods rely on several assumptions (1-4), but may not be practical for highly correlated features. How will privacy guarantees or support recovery degrade when these assumptions are not satisfied?
2) Experiments only focus on support recovery (i.e., identifying non-zero coefficients), but do not mention the prediction accuracy. Furthermore, the paper should address how to choose several hyperparameters and discuss the sensitivity of choosing them.

---

> ### Author Rebuttal · Authors · 2025-07-31
>
> Dear reviewer fXeC,
>
> We sincerely thank you for your constructive feedback on our paper. We address your concerns below:
>
> **Practicality of assumptions**
>
> Assumption 1 regarding the boundedness of the dataset is a common assumption in the high-dimensional statistics and DP literature (we refer to [1,2] and references therein). We also make a note in the paper that, while in practice one may not know the exact values of $b_x,b_y$, or such values might not exist, one can clip the values of $X,y$ to satisfy the assumption. Importantly, the only assumption required to ensure the privacy of the top-R method is Assumption 1, which can always be satisfied using clipping as discussed.
>
> Similarly, assumptions 2-4 regarding the boundedness of the true parameter vector, the Sparse Riesz Condition (SRC), and the growth of the sparsity parameter, are standard assumptions often found in the sparse linear regression literature, and we offer numerous citations in section 3 of the paper.
>
> **Experiments on prediction accuracy**
>
> In an effort to compare prediction accuracy across methods, we did a 70/30 random train test/split and implemented Algorithm 2 in [3] on the train data, wherein we use half the privacy budget (i.e., $\epsilon/2$) for variable selection using the top-R, mistakes, Samp-Agg, or MCMC methods, and the remaining half of the budget for model optimization using objective perturbation (Algorithm 1 in [3]) to obtain the regression coefficients under the privacy budget. We ran experiments with $p=100, s=5, \epsilon = 2, SNR= 5, \rho = 0.1$, and ran 10 independent trials for each value of $n$, where, for each trial, we drew 50 times from the distribution corresponding to each algorithm, and for MCMC, we used 50 independent Markov chains for each trial. After running objective perturbation over the supports, we thus had 500 distinct coefficient vectors for each method and for each $n$, and we computed the average MSE on the test data for every method. Our choices of $\lambda=120$ and MCMC iterations = 1000 were chosen to make the runtimes comparable, as done in our paper. We summarize our results below, showing that our top-R and mistakes methods outperform the competitor algorithms in terms of prediction accuracy for $n$ large enough.
>
> Average Test MSE:
>
> |                             | $n=3000$ | $n=4000$ | $n=5000$ | $n=6000$ |
> |-----------------------------|-----------|-----------|-----------|-----------|
> | top-R                       | 1.894    |  0.650    | 0.290    | 0.276    |
> | mistakes                    | 1.064    | 0.408    | 0.300    | 0.275    |
> | Samp-Agg                    | 1.490   | 0.980    | 0.677    | 0.480    |
> | MCMC                        | 0.532    | 0.386    | 0.329   | 0.317   |
>
>
> **Ablation study to choose hyperparameters**
>
> For more details on how the choice of R was selected for our experiments, we provide additional information in section B of the appendix. In summary, choosing a larger value of R improves the support recovery quality, but increases the computational cost of the method. Our choice of $R=2+(p-s)s$ is designed to encompass the optimal solution, the solutions with the support differing from the optimal by one, and at least a solution with two differences in the support relative to the optimal. This choice seems to result in reasonable runtime, while leading to good model selection accuracy.
>
> In addition, we have performed an ablation study to show the effect of changing $R, \lambda$, and $(b_x, b_y)$ on the fraction of correct supports as well as F1 score, fixing $n=3500, p=100, s=5,  \epsilon = 1, SNR= 5$, and $\rho = 0.1$. We ran 10 independent trials and drew 100 times from the distribution corresponding to our algorithms.
>
> - Ablation study of $R$: We present our results below for $R$. Note we used $\lambda=120$ and $(b_x, b_y) = (0.5,0.5)$. As intuitively expected and shown formally in section A.3 in the appendix, increasing $R$ has a positive effect on support recovery.
>
> | | $R = 177$ | $R = 277$ | $R = 377$ | $R = 477 ~(\text{i.e.}  ~ 2+(p-s)s)$ |
> |-|------------|-----------|-----------|-----------|
> |Average # correct supports / number of trials|0|0.0025|0.005|0.57
> |Average F1 score|0.047|0.049|0.0535|0.59
>
> - Ablation study of $\lambda$: We present our results below for tuning $\lambda$. Note we used $R=2+(p-s)s$ and $(b_x, b_y) = (0.5,0.5)$.
> We observe that choosing a very large $\lambda$ can have a negative effect on support recovery. However, increasing $\lambda$ is beneficial to the runtime of our outer approximation solver. Therefore, we choose a moderate value for $\lambda$ to ensure the runtime of our method remains comparable to MCMC.
>
> Average F1 scores:
> |  |$\lambda=120$ | $\lambda=140$ | $\lambda=160$ | $\lambda=180$|
> |--|---|---|----|---|
> |top-R|0.550|0.534|0.374|0.284|
> |mistakes|0.987|0.988|0.982|0.970|
>
> Average # correct supports / number of trials:
> |  |$\lambda=120$ | $\lambda=140$ | $\lambda=160$ | $\lambda=180$|
> |--|---|---|----|---|
> |top-R|0.519|0.501|0.328|0.236|
> |mistakes|0.938|0.941|0.910|0.852|
>
>
> - Ablation study of $(b_x,b_y)$: For simplicity, we set $b_x = b_y$. Note we used $R=2+(p-s)s$ and $\lambda = 300$. We present the results below. Our results imply that choosing the clipping constants to be too small or large can have a negative effect on the support recovery quality. In practice, one can tune these hyper parameters via cross-validation.
>
>
> Average F1 scores:
> |  |$b_x=0.5$ | $b_x=1$ | $b_x=1.5$ | $b_x=2$|
> |--|---|---|----|---|
> |top-R|0.111|0.3915|0.0905|0.0505|
> |mistakes|0.952|0.9795|0.9445|0.8065|
>
> Average # correct supports / number of trials:
> |  |$\lambda=120$ | $\lambda=140$ | $\lambda=160$ | $\lambda=180$|
> |--|---|---|----|---|
> |top-R|0.0625|0.35|0.04|0|
> |mistakes|0.7675|0.9|0.735|0.2575|
>
>
> References:
>
> [1] Roy, S., Wang, Z., & Tewari, A. (2024). On the computational complexity of private high-dimensional model selection via the exponential mechanism. In Advances in Neural Information Processing Systems (NeurIPS).
>
> [2] Chen, L. Y., & Lee, S. (2023). Sparse quantile regression. Journal of Econometrics, 235(2), 2195–2217.
>
> [3] Kifer, D., Smith, A., & Thakurta, A. (2012). Private convex empirical risk minimization and high-dimensional regression. In Proceedings of the Conference on Learning Theory (COLT), JMLR Workshop and Conference Proceedings.

---

> ### Author Response · Authors · 2025-08-01
>
> We apologize for a typo in the last table, the columns represent the different values of $(b_x,b_y)$ that were tested, not $\lambda$ values. The correct table is shown below.
>
> Average # correct supports / number of trials:
> |  |$b_x=0.5$ | $b_x=1$ | $b_x=1.5$ | $b_x=2$|
> |--|---|---|----|---|
> |top-R|0.0625|0.35|0.04|0|
> |mistakes|0.7675|0.9|0.735|0.2575|

---

### Note · Authors · 2025-08-13

We thank the reviewers for their thoughtful discussion. In our rebuttal, we added new experiments and clarifications that directly address the main requests:

**Prediction accuracy across methods**. We implemented objective perturbation, splitting the privacy budget evenly between feature selection and model fitting, and reported the average test MSE for $p=100$ over multiple values of $n$. The top-R and mistakes methods outperform other methods for sufficiently large $n$ at comparable runtimes. We also measured utility loss (objective gap to the non-private BSS optimum) under the same setup, finding the same trend.

**Ablations to choose hyperparameters**. We performed ablations on $R$, $\lambda$, and clipping bounds $(b_x,b_y)$ to map accuracy–runtime trade-offs. For example, very large $\lambda$ can reduce support recovery, while moderate values improve runtime. The bounds $(b_x,b_y)$ should be tuned (e.g., via CV) rather than set too small or large. Our chosen $R$ value covers supports close to optimal in Hamming distance while keeping runtimes reasonable.

**Privacy assumptions**. The discussion converged on a framing: the top-R method provides unconditional $(\varepsilon',0)$-DP (under boundedness), improving on prior work; the mistakes method attains $(\varepsilon,0)$-DP under an additional high-probability separation assumption similar to methods in prior work dependent on additional assumptions for privacy guarantees [1,2].

**Larger-$s$ regime**. To show performance when sparsity is lower, we ran experiments with $p=100$ and $s=20$. As $n$ grows, the mistakes method overtakes Samp-Agg and MCMC in both fraction of correct supports and F1 score.

**Scope and generality**. Our MIP outer approximation extends to any convex loss (e.g., Huber, quantile) without changes to the algorithmic scaffold. Our theoretical support recovery results focus on least squares, aligning with DP sparse selection literature.

**Presentation**. We will correct typos, adjust the appendix to fit page margins, and expand the conclusion and related work section.

Reviewers responded positively: one indicated an intention to raise their score, another supported acceptance, and another noted that our responses addressed several questions. We replied point-by-point to all comments; for one reviewer, acknowledgments have not yet appeared in the thread, but the corresponding experiments and clarifications are posted.

[1] Roy et al., NeurIPS 2024.

[2] Thakurta & Smith, COLT 2013.

---

### Decision · Program_Chairs · 2025-09-17

**Decision:**

Accept (poster)

**Comment:**

In this paper the authors propose new algorithms for differentially-private variable selection. Their methods are based on the exponential mechanism but incorporate novel modifications to avoid exhaustive calculation. After discussion, all of the reviewers felt positively about this paper. In particular, the problem is interesting and unique, there are compelling theoretical insights (particularly the technique for saving computation in the exponential mechanism), the methods appear to perform well empirically, and the paper is clear and well-written. The authors are encouraged to incorporate the reviewers' comments in their final revision (along with the new experiments they shared in their rebuttals).